# Seasonal deposition processes and chronology of a varved Holocene lake sediment record from Lake Chatyr Kol (Kyrgyz Republic)

Julia Kalanke[1], Jens Mingram[1], Stefan Lauterbach[2], Ryskul Usubaliev[3], Rik Tjallingii[1], Achim Brauer[1,4]

[1]GFZ German Research Centre for Geosciences, Section 'Climate Dynamics and Landscape Evolution', Potsdam, Germany
[2]University of Kiel, Leibniz Laboratory for Radiometric Dating and Stable Isotope Research, Kiel, Germany
[3]Central Asian Institute for Applied Geoscience, Bishkek, Kyrgyzstan
[4]University of Potsdam, Institute of Geoscience, Potsdam, Germany

*Correspondence to*:        Julia Kalanke (juliak@gfz-potsdam.de)
10                          Telegrafenberg C322
                            14473 Potsdam/Germany
                            +49 331 288-1379

**Abstract.**

Microfacies analysis of a sediment record from Lake Chatyr Kol (Kyrgyz Republic) reveals the presence of seasonal laminae
(varves) from the sediment base dated at $11,619 \pm 603$ years Before Present (BP) up to ~$360 \pm 40$ years BP. The Chatvd19 floating varve chronology relies on replicate varve counts on overlapping petrographic thin sections with an uncertainty of $\pm$ 5 %. The uppermost non-varved interval was chronologically constrained by $^{210}$Pb and $^{137}$Cs γ-spectrometry and interpolation based on varve thickness measurements of adjacent varved intervals with an assumed maximum uncertainty of 10 %. Six varve types were distinguished, are described in detail and show a changing predominance of clastic-organic, clastic-calcitic or -
aragonitic, calcitic-clastic, organic-clastic and clastic-diatom varves throughout the Holocene. Variations in varve thickness and the number and composition of seasonal sublayers are attributed to 1) changes in the amount of summer or winter/spring precipitation affecting local runoff and erosion and/or to 2) evaporative conditions during summer. Radiocarbon dating of bulk organic matter, daphnia remains, aquatic plant remains and *Ruppia maritima* seeds reveal reservoir ages with a clear decreasing trend up core from ~6,150 years in the early Holocene, to ~3,000 years in the mid-Holocene, to ~1,000 years and less in the
late Holocene and modern times. In contrast, two radiocarbon dates from terrestrial plant remains are in good agreement with the varve-based chronology.

## 1 Introduction

The interplay of the large atmospheric circulation systems in Central Asia (CA), including the Siberian High, the Westerlies
and the Indian Monsoon and their influences on regional climate is still not fully understood. This is partly due to the large contrasts of landscapes (high mountains, deep basins, large water bodies and deserts), the low spatial and temporal coverage

of high-resolution paleo-climate archives and the partly problematic dating of these archives in this area. Information about Holocene climate variability in CA derive from several types of archives, including tree rings (Esper et al., 2003), speleothems (Fohlmeister et al., 2017; Wolff et al., 2017), ice cores (Aizen, 2004), aeolian deposits (Huayu et al., 2010) and lakes (Heinecke et al., 2017; Lauterbach et al., 2014; Mathis et al., 2014; Rasmussen et al., 2000; Ricketts et al., 2001; Schwarz et al., 2017). However, none of these lake records has reported annually laminated sediments. In Kyrgyzstan, varves have only been reported from Lake Sary Chelek for the short time interval from ~1940´s to 2013 (Lauterbach et al., 2019). Other varved records in the wider region are from Lake Telmen in northern Mongolia which includes discontinuous varved intervals during the last ca. 4,390 cal years BP (Peck, 2002) and from Lake Sugan in north western China covering the last ~2,670 years BP (Zhou et al., 2007). Deciphering Holocene climate changes based on limnic records in CA is challenging due to the influences of several factors: 1) chronological uncertainties caused by the scarcity of datable terrestrial plant material at high altitudes and often large $^{14}$C reservoir effects of aquatic organic material (Hou et al., 2012; Lockot et al., 2016; Mischke et al., 2013), 2) human influence (Boomer et al., 2000; Mathis et al., 2014), possibly overprinting the natural climate signals in the archives and 3) variations in the dominance of the mid-latitude Westerlies, the Siberian High and the Asian Monsoon system leading to different spatial and temporal climate effects over CA (Chen et al., 2008; Herzschuh, 2006; Mischke et al., 2017; Schroeter et al., 2020). All these factors can hamper data comparison and may lead to different paleo-environmental interpretations (Chen et al., 2008; Hou et al., 2012; Mischke et al., 2017). The investigation of varved lake sediments offers the unique opportunity for independent dating through varve counting. In addition, the description of varve micro-facies has the potential to provide detailed insights into environmental and climate variations at a seasonal scale. The sediment record from Lake Chatyr Kol is the first varved record from CA covering most of the Holocene and the main goal of this study is to establish a robust age model through an integrated dating approach primarily based on varve counting. Varve counting requires an in-depth understanding of seasonal deposition of all varve types occurring in the sediment record. Therefore, we apply continuous microfacies analysis for the entire sediment profile to describe the Holocene evolution of varve formation in detail and discuss fundamental deposition processes.

## 2. Study site

Lake Chatyr Kol (40°36' N, 75°14' E) (Fig.1) is located at ~3,530 m above sea level (a.s.l.) in the intramontane Aksai Basin (De Grave et al., 2011; Koppes et al., 2008) in the southern Kyrgyz Republic. In the north, the basin is restricted by the At Bashy Range and in the south by the Torugat Range resulting in a catchment area of about 1,084 km$^2$. Geologically, the surrounding mountain ranges belong to Silurian to Carboniferous sedimentary-volcanogenic complexes of marine-continental collision zones, consisting of limestones and dolomites, that crop out directly along the northern lake shore, as well as siliceous rocks, shales and scattered Permian granites that crop out in the south and north-east (Academy of Science of the Kyrgyz SSR, 1987). The modern lake, which has a maximum length of 23 km, a width of 10 km and a maximum depth of 20 m in its

western-central part, is endorheic and separated from the neighbouring Arpa river basin in the north-west by a moraine (Shnitnikov, 1978). The moraine originates from glacial advances of unknown age from the western Torugat mountain range. Present day glaciers exist above ~4000 m a.s.l. on the Torugat and At Bashy mountain ranges but only some of the At Bashy glaciers drain into Chatyr Kol via the Kegagyr River. The lake further receives convective rainfall in summer (Aizen et al., 2001). A shallow dam at ~ 3,550 m a.s.l. hinders outflow to the east. Modern climate conditions are generally dry and mainly controlled by the Westerlies and the Siberian Anticyclone Circulation (Aizen et al., 2001; Koppes et al., 2008). Mean annual precipitation is ~275 mm/a as indicated by Aizen´s (2001) evaluation and spatial averaging of annual precipitation of historical records published by Hydrometeo (Reference Book of Climate USSR, Kyrgyz SSR, 1988). This is comparable to long-term instrumental data from nearby (about 50 km away) weather stations at comparable altitudes, where annual precipitation means are 237 mm/a (station "Chatirkul", 75°8´E, 40°6´N, 3,540 m a.s.l, AD 1961–1990) and 294 mm/a (station "TienShan", 78°2´E, 41°9´N, 3,614 m a.s.l., AD 1930–2000) (Williams and Konovalov, 2008). Monthly mean temperatures range from -26.0 to 8.0 °C (Koppes et al., 2008; Academy of Science of the Kyrgyz SSR, 1987) with means of -5.4 °C (station "Chatirkul") and -7.6 °C (station "TienShan") (Williams and Konovalov, 2008). The salinity of the lake water ranges from 1.06–1.15 g/l in the deeper western part of the lake to 0.24 g/l in the shallower eastern part near the inflow (Romanovsky, 2007). Measurements of oxygen concentrations (YSI Pro 6600 V2) during a field trip in July 2012 ranged from ~6 mg/l at the water surface to ~1 mg/l in 19 m depth with a clear oxygen minimum zone below ~11 m depth (Fig. 2). Water surface and bottom water temperatures (YSI Castaway CTD) at 19 m depth reached 13.2 °C and 9.4 °C. Specific conductivity (CTD) ranged from 1,902 µS/cm$^{-1}$ at 19 m to 1,825 µS/cm$^{-1}$, pH was ~9 (YSI Pro 6600 V2). Secchi depth was about 4 m. Nowadays, the lake is largely occupied by the amphipod *Gammarus alius* sp. nov. (Sidorov, 2012) and no fish live in the lake (Shnitnikov, 1978).

The permafrost level is located at a depth of 2.5-3 m in the littoral coast zones and the lake is covered by ice from October to April (Shnitnikov, 1978). Modern permafrost thawing results in unstable shores visible at the Maloye lake located < 2 km to the South of Chatyr Kol (Fig. 1 Photo) and the development of small ponds on the shallow south-western shore of this lake and lake Chatyr Kol during summer. Several terraces in the north, south and east of the lake result from Pleistocene-Holocene lake level fluctuations (Romanovsky and Shatravin, 2007; Shnitnikov, 1978). Vegetation around the lake is generally poor and represented by high alpine meadows (Shnitnikov, 1978; Taft et al., 2011).

## 3. Methods

### 3.1 Coring and Composite profile

Five parallel cores each of 3 to 6 m length have been retrieved in 2012 from the deepest part of the lake (40°36.37' N, 75°14.02' E) by using an UWITEC piston corer (Fig. 1, Tab. 1). All cores were opened, split and photographed at GFZ Potsdam, where they are archived in a cold store at 4°C. A continuous composite profile of 623.5 cm length (CHAT12) was established by correlating the individual, overlapping cores via macroscopically visible marker layers (Fig. 3). Furthermore, seven parallel

gravity cores (SC17_1-7) have been retrieved with a UWITEC gravity corer in 2017 (Fig.1, Tab.1) to recover the undisturbed sediment-water interface, from which the best preserved parallel core SC17_7 was used for gamma spectrometric analysis.

## 3.2 Sediment microfacies analysis and varve counting

Continuous 10-cm-long sediment slabs with an overlap of 2 cm were taken from the whole composite profile to prepare large-scale petrographic thin sections. Thin section preparation followed the method described by Brauer and Casanova (2001) and included freeze-drying and vacuum impregnation of the sediment slabs with Araldite epoxy resin. Microfacies analysis, including a semi-quantitative evaluation of planktic and periphytic diatoms, aquatic plant remains (e.g. *Potamogeton* sp., *Ruppia maritima*), ostracods, daphnia, characeae and chrysophytes, was carried out on a Zeiss Axioplan microscope using different magnifications (25–400 x) and included measurement of varve thicknesses, microfacies/varve type characterization, the definition of varve boundaries and the development of process-related deposition models. A varve quality index (VQI) ranging from 0-5 was given for each varve, comparable to the method from (Żarczyński et al., 2018) and references therein.

- VQI 0 = no varves or strongly disturbed varved sequences, no reliable counting (interpolation)
- VQI 1 = very low varve preservation, horizontally discontinuous varve and less well-preserved sublayer boundaries, difficult counting
- VQI 2 = low varve preservation, occasional horizontally discontinuous varve and sublayer boundaries, reliable counting
- VQI 3= medium varve preservation, horizontally continuous varve and sublayer boundaries, only small disturbances, reliable counting
- VQI 4= high varve preservation, clearly distinguishable varve and sublayer boundaries, reliable counting
- VQI 5= highest varve preservation, clearly distinguishable varve and sublayer boundaries, no disturbances, reliable counting

Varve counting was performed to establish a floating varve chronology. Non-varved intervals (VQI=0) between varved sediment sections were therefore interpolated by using the mean of sedimentation rates derived from about 20 varves above and below the non-varved part. Varves were counted twice by the same author. Counting uncertainty estimates were first assessed by the percentage deviation of the second to the first count within one thin section. The mean of these deviations was used as an overall counting uncertainty estimate and assigned to the entire varved record. The uncertainty estimates were thus also assigned to interpolated sequences.

## 3.3 XRF element mapping

X-ray Fluorescence (XRF) element mapping was performed on two selected Araldite impregnated sediment blocks (ca. 2 x 10 cm), which were prepared for thin sections used for the microfacies analyses. XRF element mapping of these two sediment blocks allows linking micro-facies analyses of typical varve types directly with geochemical sediment compositions. Element mapping was performed at 50 µm resolution and covering most of the surface of the sediment block (15 x 100 mm) using a Bruker M4 Tornado at GFZ Potsdam. This scanner is equipped with a Rh X-ray source operated at 50 kV and 600 µA in

combination with poly-capillary X-ray optics that irradiate a spot of 20 µm for 50 ms. After measuring and an initial spectrum deconvolution, normalized element intensities are used to visualize relative element abundances as 2D maps.

### 3.4 Radiometric dating

### 3.4.1 Radiocarbon dating

In total, 36 accelerator mass spectrometry (AMS) [14]C measurements were carried out at the Pozńan Radiocarbon Laboratory
in Poland. Samples for [14]C measurements comprised two pieces of wood, bulk TOC samples, aquatic plant macro remains, daphnia remains and *Ruppia maritima* seeds (Tab.2). Additional samples of recent living daphnia and aquatic plants have been collected to assess the modern [14]C reservoir effect. The resulting conventional [14]C ages were calibrated using OxCal 4.3 (Ramsey, 2009) with the IntCal13 calibration curve (Reimer et al., 2013).

### 3.4.2 Gamma spectrometry dating

Gamma spectrometry measurements were performed on 0.5-cm-thick sediment slices that were continuously sampled from the upper 15.0 cm of gravity core SC17_7 (Suppl. Table. 1). The samples were freeze-dried and sieved through a 200 µm mesh for homogenization and removal of larger plant particles. Individual sample-aliquots were filled into gas-tight sealable low-activity Kryal© tubes at identical fill heights and accurately weighted. After sufficient in-growth-time, the gamma energies of $^{210}$Pb ($T_{1/2}$= 22 a) and $^{214}$Pb ($T_{1/2}$= 26.8 min), which is a daughter nuclide of $^{222}$Rn (T1/2= 3.8 d), were measured at 46.54,
295.24 and 351.93 keV. In addition, the gamma energies of $^{137}$Cs ($T_{1/2}$= 30.1 a) were measured at 661.66 keV. For this purpose, the Kryal© tubes were placed into shielded measurement chambers equipped with two well-type germanium detectors G1 and G2 (Canberra Industries) for ~1.5 to 7 days at GFZ Potsdam (Suppl. Tab. 1) (Schettler et al., 2006). Hardware control, data storage, and spectrum analysis were realized with the software Genie 2000 (Canberra Industries). The average counting uncertainty for $^{210}$Pb was 5.9 %, for $^{214}$Pb 7.7 % (295 keV) and 3.7 % (351 keV) and for $^{137}$Cs 5.2 %. Efficiency calibrations
were carried out for $^{210}$Pb, $^{214}$Pb and $^{137}$Cs with the same analytical setup using a lab-internal standard and the "Loess Nussloch" standard (Potts et al., 2003). Blank activities for $^{137}$Cs were negligible while average $^{210}$Pb blank activities of 10 mBq/g for detector G2 and $^{214}$Pb blank activities of 9 mBq/g for the detectors G1 and G2 were considered. The activity measurements of $^{214}$Pb were used to quantify the proportion of supported $^{210}$Pb ($^{210}$Pb$_{supp}$) produced by the decay of $^{226}$Ra in the sediment. The activity of unsupported $^{210}$Pb ($^{210}$Pb$_{unsupp}$) in the sediment, which originates from the decay of $^{222}$Rn in the atmosphere and
associated aeolian deposition, is quantified by the difference between measured $^{210}$Pb$_{total}$ and $^{210}$Pb$_{supp}$. We selected sections that showed linear correlations in the semi-logarithmic plot of $^{210}$Pb$_{unsupp}$ versus depth to infer average sedimentation rates using the constant initial concentration (CIC) model (c.f. Appleby, 2002) (Suppl. Tab. 2). Intercalated sediment sections showed nearly uncorrelated ln($^{210}$Pb$_{unsupp}$) vs. depth relationships at 10.25–9.25, 6.25–4.25 and 2.25–1.75 cm depth (Suppl. Fig.2). Therefore, the initial $^{210}$Pb$_{unsupp}$ activities of samples that bridged these sections were used alternatively to determine
time intervals between these samples to infer a chronology. To assess possible changes of the sedimentation regime we

additionally calculated sedimentation rates of each 0.5-cm-thick sediment slice using the CRS model (constant initial $^{210}Pb_{unsupp}$ supply) (c.f. Appleby, 2002; Appleby and Oldfield, 1978) (Suppl. Tab.3).

## 4. Results

### 4.1 Lithology

The composite profile can be subdivided into six lithological units (Fig. 3). Lithozone (LZ) I from 623.5 to 566.0 cm depth consists of greyish-brownish clastic-calcareous sediments. It shows mm-scale laminations of fine sandy and silty to clayey layers. LZ II (566.0–480.0 cm) exhibits intercalations between horizons of very fine, mm-scale laminated brownish-reddish organic-rich sediments and sections with greyish calcareous sediments. Brownish-reddish intercalating horizons of mm-scale laminated organic and calcareous sediments characterize LZ III from 480.0 to 273.0 cm depth. LZ IV (273.0–130.0 cm) is

characterized by brownish-reddish mm-scale laminated organic-rich sediments with intercalated horizons rich in aquatic plant remains, which occur at 232.0–223.0 cm, 185.0–180.0 cm and 164.0–130.0 cm depth. LZ V (130.0–41.0 cm depth) starts with a 16-cm-thick interval of dark grey mm-scale laminated calcareous sediments, followed by brown mm- to cm-scale laminated sediments until 41.0 cm depth. Laminations are only poorly preserved between 63.0 and 41.0 cm depth. The uppermost sediments of LZ VI (41.0–0.0 cm depth) consist of homogenous, brownish-greyish calcareous sediments, which are rich in

aquatic plant remains. The uppermost centimeter is enriched in calcite and exhibits greyish faint laminations.

### 4.2 Sediment microfacies analysis

Microscopic sediment analysis revealed that clastic sublayers are present throughout the finely laminated sediments below 63.0 cm depth (Fig. 4.1). These clastic sublayers are variably intercalated with calcitic, aragonitic and organic sublayers and thus form different types of cyclic successions. In total, we classified six different types of sublayer successions as described

below. The name for these types reflects the dominant sublayer for each of the six types. For example, the 'clastic-organic type' is characterized by the dominance of clastic sublayers, while in the organic-clastic type organic sublayers dominate. The names are not related to the order of sublayer succession within each type. Changing dominances of different sublayer successions reflect the lithozones.

#### 4.2.1 Clastic-aragonitic type

Clastic-aragonitic laminae are rare (2.7 %), mainly occurring in LZ I and particularly at 600.0-605.0, and 609.0-616.0 cm composite depth. This subtype is composed of three sublayers and the mean thickness is 0.59 mm (Fig. 4.1a, Suppl. Fig. 2a). These laminae exhibit the general pattern of clastic-organic laminae in LZ I, with a coarse-grained and thick basal detrital sublayer, but the overlying mixed (detrital calcite, mica, fsp, qtz and medium amounts of endogenic calcite) fine-grained sublayer additionally contains idiomorphic aragonite needles that are not found in clastic-organic varves. The sublayer

succession ends with an amorphous organic matter sublayer.

### 4.2.2 Calcitic-clastic type

The deposition of calcitic-clastic laminae (6 %) with a dominating endogenic calcite sublayer is restricted to LZ II. This subtype is composed of three sublayers and the mean thickness is 0.41 mm, with a maximum of 2.0 mm. Calcitic–clastic laminae (Fig. 4.1b lower part, Suppl. Fig. 2b) are usually characterized by a basal detrital sublayer which, however, is not developed in all calcitic-clastic laminae. The overlying sublayer generally exhibits low species abundances of diatom frustules, chrysophyte cysts, aquatic plant remains, daphnia, ostracods and characeae but massive and fine-grained endogenic calcite, which is not the case in the clastic-calcitic laminae subtype. Endogenic calcite formed in the water column (Fig. 4.2a) is recognized by its well-developed idiomorphic rhombohedral shapes. Scattered detrital grains occasionally occur within the endogenic calcite matrix. One depositional cycle ends with an amorphous organic matter sublayer.

### 4.2.3 Clastic-diatom type

Clastic-diatom laminae (20 %) occur in LZ II, III and IV. This subtype is composed of three sublayers and the mean thickness vary between 0.28 mm (LZ II), 0.34 mm (LZ III) and 0.35 mm (LZ IV). The depositional cycle starts with a basal detrital sublayer, which is overlain by a finer-grained mixed sublayer (detrital calcite, mica, fsp, qtz) occasionally containing chrysophytes and different diatom taxa. The third sublayer is formed by diatom blooms exclusively consisting of the planktic diatom species *Cyclotella choctawhatcheeana* (pers. comm. Anja Schwarz, TU Braunschweig) (Fig. 4.1b upper part, Suppl. Fig. 2c).

### 4.2.4 Clastic-calcitic type

The second most common lamina subtype (23.5 %) are clastic-calcitic laminae (Fig. 4.1c lower part, Suppl. Fig. 2d), which are most abundant in LZ I, II, III and V. This subtype is composed of four to five sublayers and the mean total thickness varies between 0.95 mm (LZ I), 0.35 mm (LZ II), 0.72 mm (LZ III), 1.56 mm (LZ IV) and the maximum value of 5.0 mm in LZ V. Clastic-calcitic laminae exhibit a basal detrital sublayer with a sharp lower boundary, which is followed by a bloom layer of chrysophytes and/or diatoms, occurring sporadically after and/or within the detrital sublayer. The third, overlying mixed sublayer contains medium amounts of endogenic as well as fine-grained detrital calcite (Fig. 4.3a), as well as mica, fsp and qtz grains but low amounts of diatom frustules and chrysophyte cysts. One depositional cycle typically ends with an amorphous organic matter sublayer. In LZ V, these clastic-calcitic laminae occasionally contain a very fine-grained, light greyish, micritic sublayer before the cycle ends with the amorphous organic sublayer.

### 4.2.5 Organic-clastic type

Horizons of organic-clastic laminae (5.2 %) with dominating organic sublayers are mainly present within LZ IV and V (Fig. 4.1d, Suppl. Fig. 2e) particular at 261.0-252.0, 176.0-173.0, 150.0-126.0 and 122.0-110.0 cm depth. This subtype is composed of three sublayers and the mean thickness is 0.49 mm (LZ IV) and 1.64 mm (LZ V) with a maximum of 9 mm in LZ V.

Organic-clastic laminae exhibit an often horizontally discontinuous basal detrital sublayer (lens-shaped) in LZ IV, which is overlain by a mixed sublayer that contains detrital calcite, mica, fsp and qtz grains and many aquatic plant remains and periphytic diatoms (*Achnanthes brevipes,* pers. comm. Anja Schwarz, TU Braunschweig), whose colony chains are often preserved. One deposition cyclic ends with a yellowish amorphous organic matter layer.

### 4.2.6 Clastic-organic type

Clastic-organic laminae are present in all lithozones and most abundant in the record (42.5 % of all observed and measured laminae). This microfacies type is composed of four sublayers of which the most prominent is a basal clastic-detrital sublayer with a sharp lower boundary. The basal detrital sublayer contains mainly detrital calcite, which is distinguished from endogenic calcite by microscopic analyses. Detrital calcite is characterized by irregularly shaped gains and generally larger grain sizes of average of 0.6 mm and up to 1.82 mm in LZ I. Detrital layers further contain siliciclastic minerals such as mica, quartz (qtz) and feldspars (fsp). This basal layer is often, but not regularly overlain by chrysophyte and/or diatom blooms and a third, mixed sublayer containing mainly fine-grained detrital calcite, mica, fsp and qtz with low amounts of endogenic calcite and varying amounts of diatom frustules, chrysophytes, characeae, ostracods and daphnia. The deposition cycle ends with a yellowish layer of amorphous organic material.

The mean thickness of clastic-organic laminae differs between the lithozones. In LZ I, the mean thickness is 0.59 mm with a maximum thickness of 3.1 mm. In LZ I, the basal detrital sublayer is thick and coarse-grained, rich in pyrite, and contains mainly silt-to fine sand-sized grains and occasionally sand-sized qtz, calcite and fsp grains, whereas diatoms and chrysophytes are rare. In LZ II and III, clastic-organic laminae are less thick with a mean thickness of 0.27 mm and 0.48 mm respectively. In these lithozones, the basal sublayer contains no sand-sized particles. In LZ IV, mean varve thickness is 0.43 mm and the basal sublayer is often lens-shaped and horizontally discontinuous. In LZ V, between 130.0 and 63.0 cm depth, thickest clastic-organic laminae occur with a mean thickness of 1.5 mm and a maximum thickness of up to 7.0 mm (Fig. 4.1e, Suppl. Fig. 2f). These clastic-organic laminae often include an additional detrital sublayer intercalated in the finer grained mixed sublayer.

### 4.2.7 Homogenous sediments

The uppermost 41.0 cm of the sediment record consist of homogenous sediments, containing a fine-grained mix of autochthonous and allochthonous calcite, mica, qtz and fsp. The sediments are generally rich in organic remains, such as aquatic plant remains, chrysophytes, diatoms and chlorophytes (*Botryococcus*). Faint and discontinuous calcite laminae occur in the uppermost centimeter (Fig. 4.1f).

### 4.3 XRF element mapping

The two selected impregnated sediment blocks from 507 to 497.5 cm (XRF-Map 1 Fig. 4.2) and from 346.5 to 338.5 cm depth (XRF-Map 2 Fig. 4.3) contain calcitic-clastic, clastic-diatom, clastic-calcitic and clastic-organic microfacies types. These sediments are dominated by alternating calcitic and siliciclastic sediments represented by the elements Ca, Sr, Mg, Si and Al,

respectively (Fig. 4.2 and 4.3). Color variations of the element maps show that the calcitic and siliciclastic sediments are clearly separated in the XRF-Map 1 sample (Fig. 4.2) but slightly more mixed in the XRF-Map 2 (Fig. 4.3). In both XRF-Maps 1 and 2, the carbonate sublayers are enriched in Sr (Fig. 4.2 and Fig. 4.3), whereas in the XRF-Map 1 the additional enrichment of

Mg (Fig. 4.2) indicates the presence of Sr- and Mg-rich carbonates. Microfacies analysis shows that Mg- and Sr- rich calcite sublayers in XRF-Map 1 are predominantly of endogenic origin (Fig. 4.2a), whereas the Sr- rich calcite layers in XRF-Map 2 contain mixed endogenic and resuspended calcites (Fig. 4.3a). Moreover, detrital carbonates occur predominantly as individual grains in the siliciclastic sediments and sublayers and are shown by Ca and Mg in the XRF element maps (Fig. 4.2 b and Fig. 4.3 b). Siliciclastic muds of clastic-organic and clastic-diatom varves are represented by the co-occurrence of Al and Si in the

XRF element maps, whereas Al is absent in diatomaceous sublayers (Fig. 4.2 and 4.3).

### 4.4 Chronology

#### 4.4.1 Floating varve chronology (623.5 - 63.0 cm)

A floating varve chronology labelled as Chatvd19 (Fig. 5b) was established for the composite profile below 63.0 cm depth and comprises a total of 11,259 counted and interpolated varves. Based on the interpretation of laminations as varves, 9,026 of the

total 11,259 varves were counted which is equal to 80.2 %. The first varve count reveals 9,026 varves and is the base for the floating varve chronology. Although the total varve number of 8,955 obtained by the second count is very similar to the first count, larger deviations between the two varve counts in individual sediment sections occur throughout the sediment record due to varying stages of varve preservation as expressed in the VQI (Fig. 5a). The largest deviations occur in LZ I (603.0-595.0 cm) with 23.7 %, in LZ II (490.0-484.0 cm) with ~13 %, in LZ III (413.0-405.0 cm) with ~16 %, in LZ IV (141.0-134.0

cm depth) with 19.5 % and in LZ V (65.0-63.0 cm depth) with 7.7 %. The lowest deviations (<1 %) were obtained in LZ II at 539.0-530.0 cm and 498.0-490.0 cm, in LZ III at 451.0-445.0 cm, 421.0-413.0 cm, 375.0-369.0 cm, 299.0-290.0 cm and 282.0-275.0 cm, in LZ IV at 197.0-190.0 cm and in LZ V at 125.0-119.0 cm, 116.0-113.0 cm and 73.0-65.0 cm depth. Interpolated sequences are unevenly distributed within the record and are mainly present within LZ IV. The longest interpolated sequences occur in LZ IV with ~5 cm from 198.0-193.0 cm depth and in LZ III with almost 7 cm between 444.0-437.0 cm depth. A VQI

(Fig.5a) of 1 is represented by 5.6 % of the total varves, VQI 2 by 8.4 %, VQI 3 by 26.4 %, VQI 4 by 18.3 % and VQI 5 by 21.4 %. The calculated mean deviation between the two varve counts of ~5 % (Fig. 5a) is used as a conservative uncertainty for the floating varve chronology to consider high uncertainties in individual sediment sections in a more realistic way, despite the similar total number of varves counted. The floating varve chronology has a basal age of $11619 \pm 603$ years BP.

#### 4.4.2 Chronology of the non-varved uppermost sediments

The uppermost 63.0 cm of the sediment profile are not varved and thus require alternative dating approaches including [210]Pb dating, activity profiles of [137]Cs and sedimentation-rate based interpolation. First, we measured [210]Pb activity concentrations of the uppermost 15 cm of short core SC17_7 and applied the CIC and CRS models (Fig. 5c & 6, Suppl. Tab. 3). SC17_7 is

correlated to the composite profile through macroscopically visible facies change at 1.0 cm composite depth and through a laminated section from 45.0-41.0 cm composite depth (Suppl. Fig.1). The CIC and CRS-model based chronologies are broadly

consistent and particularly date sediments at 8.75 cm (SC17_7) or 7.5 cm composite depth to AD 1945/46 (Fig. 5c and Fig. 6b, Suppl. Table 3,). This coincides with the onset of increased [137]Cs activity concentrations (Fig. 5c, 6c) marking the onset of nuclear weapon testing in AD 1945 (Ferm, 2000; Kudo et al., 1998; Norris and Arkin, 1998). Therefore, we applied the date of AD 1945 (8.75 cm in core SC17_7) as an anchor point for the chronology of the uppermost 63.0 cm of the composite profile. According to micro-facies based sedimentological correlation, this anchor point is located at 7.5 cm composite depth. The

section of homogeneous sediments from this point down to 63.0 cm depth was interpolated. This interpolation is based on sedimentation rate calculations obtained by lead-210 dating and varve thickness measurements in adjacent sediment intervals. Calculations include sedimentation rates from the upper 7.5 cm (1.12 mm/yr), from 15 varves (1.9 mm/yr) between 41.0-63.0 cm depth and from 100 varves (1.66 mm/yr) below 63.0 cm depth and result in a mean SR of 1.56 mm/yr corresponding to 356 interpolated years. Adding the number of 356 interpolated years to the radiometric date of AD 1945 results in an age of

AD 1589 (360 years BP) at 63.0 cm depth. We assume a conservative uncertainty of ca. 10% as a maximum error for our interpolation. The anchor point thus has an age of $360 \pm 40$ years BP. The uncertainty of this anchor point is added to the varve counting uncertainty.

### 4.4.3 Radiocarbon dating

In total, we dated 36 samples of bulk organic carbon, daphnia remains, aquatic plant remains and *Ruppia martima* seeds. Only

two samples were terrestrial plant remains (wood fragments) and sufficiently large to be used for AMS [14]C dating (Tab. 2, Fig. 5 & 7). Except the two ages from terrestrial plant remains (Poz-54302 with $9988 \pm 203$ and Poz-63307 with $6140 \pm 137$ cal yr BP), all other ages deviate from the varve chronology between 155 years at 0.0 cm depth and 6,150 years at 585.0 cm depth (Fig. 7). We observe a general trend of decreasing deviations up core with the maximum deviation of ~6,150 years at 585.0 cm depth in LZ I. Looking at more detail, the deviations between radiocarbon and varve ages exhibit a prominent step-wise

increase particularly at the boundary between lithozone LZ IV and LZ V when it abruptly decreases from ~3,000 years to ~1,000 years. Modern aquatic plants collected during the field campaign in 2012 showed large modern reservoir ages of $330 \pm 30$ and $2425 \pm 25$ [14]C years and living daphnia yielded ages of $225 \pm 30$ [14]C years.

## 5. Discussion

### 5.1 Interpretation of fine laminations as varves

The construction of varve chronologies relies on the proof of seasonal origin of fine laminations (Brauer et al., 2014; Ojala et al., 2012; Zolitschka et al., 2015). Laminations are absent in the upper 63.0 cm of the Lake Chatyr Kol sediment core and cyclic successions of mixed-clastic laminations are only observed below this depth. Therefore, the seasonal origin of the Chatyr

Kol sediments cannot be proved through modern observation in sediment traps because no varves are formed in the present day. Instead, we applied process-related deposition models (Fig. 4.1) based on detailed micro-facies analyses obtained from

petrographic thin sections and compared our observations with varve types described in literature (Brauer, 2004; Zolitschka et al., 2015). We associate the observed successions of different types of mixed-clastic laminations with the formation of different seasonal sublayers that are known from lakes with carbonaceous catchments (Brauer and Casanova, 2001; Kelts and Hsü, 1978; Lauterbach et al., 2011; Lauterbach et al., 2019) and high-altitude glacial environments (Guyard et al., 2007; Leemann and Niessen, 1994). The observed successions of sublayers are interpreted as mixed varve types (clastic, -organic and –

endogenic) as defined by Zolitschka et al. (2015).

Varve formation at Lake Chatyr Kol is related to the high seasonality of the local climate with an ice cover during winter as well as strong annual variations of the temperature and precipitation affecting productivity, endogenic carbonate formation and local runoff. Varve preservation is promoted by the unique morphology of the deep western lake basin, where anoxic bottom water conditions can be maintained even under relatively low lake levels (Fig.2).

We interpret the annual sedimentary cycle to always start with the deposition of a basal detrital sublayer with a sharp lower boundary which results from winter/spring snow and/or glacial melt (Guyard et al., 2007; Leemann and Niessen, 1994; Zolitschka et al., 2015) after the ice break-up in ~April (Shnitnikov, 1978). Runoff with suspended sediment load is then likely directed through the Kegagyr River in the east but may also be the result of surface runoff through the activation of several widely distributed smaller tributaries in the catchment (Fig. 1).

Basal detrital sublayers are generally overlain by blooms of chrysophytes and/or diatoms within clastic-organic, organic-clastic, clastic-diatom and clastic-calcitic varve types. Chrysophytes and/or diatom blooms develop as a consequence of the available nutrients provided by runoff and spring overturn in combination with rising temperatures during the summer season (Zolitschka et al., 2015). The productive phase in calcitic-clastic varves is however reflected by calcite precipitation, which is the main carbonate phase (endogenic, detrital and resuspended) in the Chatyr Kol sediments. The formation of

endogenic calcite in Lake Chatyr Kol is controlled by: 1) photosynthesis, when high aquatic productivity lowers the concentrations of $CO_2$, increases the pH of the lake water and leads to a reduced solubility of $CO_3^{2-}$(Hodell et al., 1998; Kelts and Hsü, 1978; Zolitschka et al., 2015), 2) evaporation leading to an oversaturation of carbonate ions, 3) sufficient supply of dissolved cations either through surface runoff or groundwater inflow (Shapley et al., 2005). Changes in weathering and hydrological conditions can lead to variations in the supply of $Ca^{2+}$ and $Mg^{2+}$ ions and subsequently change the Mg/Ca ratio

of the lake water (Müller et al., 1972). The formation of aragonite requires high lake water Mg/Ca ratios (>12), whereas magnesium-calcite forms at lower Mg/Ca ratios (Kelts and Hsü, 1978; Müller et al., 1972). XRF element intensity maps do not provide quantitative results but do indicate, that Mg is abundant in the XRF map 1 from LZ II which results in the formation of endogenic Sr- and Mg-rich calcites (Fig. 4.2 a).

Aragonite precipitates, related to an evaporative concentration in summer, were only microscopically observed in the

intervals between 600.0-605.0 and 609.0-616.0 cm composite depth suggesting that Mg/Ca ratios probably remained above Mg/Ca ratios >12.

After the spring to early summer lake productivity, the deposition of a mixed sublayer consisting of silt- to clay-sized detrital grains and low to medium amounts of endogenic calcite is observed in all lamination types (Fig. 4.1, Suppl. Fig. 2). In clastic-calcitic varves, the mixed sublayers appear different and include especially resuspended calcite as evidenced also in Ca and Sr intensities (Fig. 4.1c, Fig. 4.3a, Suppl. Fig. 2d). The mixed sublayer indicates resuspension of shore material (littoral calcite) to the core location due to e.g. wind induced wave activity and weak runoff during the ice-cover free season from ~April to October (Shnitnikov, 1978).

The intercalation of discrete detrital layers within the mixed sublayer (Fig. 4.1e, Suppl. Fig. 2e), as observed in clastic-organic laminae in LZ V, indicates pulses of runoff of suspended material which may be caused by late rainfall events in summer (Aizen et al., 2001; Shnitnikov, 1978).

One annual depositional cycle usually ends with the deposition of a thin sublayer of very fine amorphous organic matter which is deposited under quiet water conditions when the lake was ice covered (Fig. 4.1). In lithozone V, an additional micritic sublayer is deposited before the amorphous organic sublayer at the end of the seasonal cycle in clastic-calcitic laminae when water turbulence is low.

## 5.2 Varve counting and chronology construction

The interpretation of different types of fine laminations allowed varve counting as a main tool for constructing the Chatyr Kol chronology largely based on incremental methods. Around 80% of the varves in the sediment record are double-counted in petrographic thin sections while the remaining part of ca 20% had to be interpolated based on sedimentation-rate estimates due to poor varve preservation. The resulting chronology comprises 11,259 years and is anchored to the absolute time scale at 63.0 cm sediment depth supported by a combination of lead-210 dating and occasional sedimentation rate measurements as described below. The resulting age-depth model is within uncertainties in good agreement with two calibrated AMS [14]C dates of wood pieces at 380.5 cm depth (6,140 ± 137 cal years BP; Poz-63307) and at 528.0 cm depth (9,988 ± 203 cal years BP; Poz-54302) (Fig. 5b, Tab. 2). The corresponding varve-based ages are 5,905 ± 320 years BP and 9,611 ± 505 years BP, respectively. As for all chronologies, uncertainties are inherent also to varve chronologies, which are commonly assessed via replicate counts (Brauer and Casanova, 2001; Lamoureux, 2001; Lotter and Lemcke, 1999; Ojala et al., 2012; Żarczyński et al., 2018; Zolitschka et al., 2015). However, there is no standard procedure on how to calculate and present the uncertainties (Ojala et al., 2012; Zolitschka et al., 2015). Commonly, mean values of replicate count differences, the difference of maximum and minimum counts or their standard deviation are reported (Brauer et al., 2014; Ojala et al., 2012; Żarczyński et al., 2018; Zolitschka et al., 2015). Despite the inevitable increase of cumulative uncertainties with age or depth, systematic uncertainties arise and are caused by changes in varve preservation, strongly and abruptly varying sedimentation rates and the challenging differentiation of varve types with complex structures (Ojala et al., 2012; Żarczyński et al., 2018; Zolitschka et al., 2015). The overall very small difference between the two counts of the Chatyr Kol varved record of only -71 varves is due to the compensating effect between over-and underestimations of varve counts throughout the record. For the floating varve

chronology, we therefore compare the results for each individual thin section comprising between a maximum of 324 (506.8-497.6 cm) and a minimum of 13 (varves) (65.4-63.0 cm) (Fig. 5a, Fig. 8).

Counting uncertainties for individual thin sections are reported as their percentage deviation from the first count used for the chronology and range between 0 and 23.7 % (Fig. 5a, Fig. 8). Largest deviations of 23.7 % in LZ I are caused by a low visibility of varve boundaries and by coring artefacts. Deviations of ~13 % in LZ II and of ~16 % in LZ III result from the abrupt intercalations between clastic-organic, clastic-diatom, clastic-calcitic and calcitic-clastic varve types with varying varve thicknesses (Fig. 4 LZ II & LZ III). Deviations in LZ IV with a maximum of 19.5 % coincide with generally lowest VQI values (Fig. 8) and result from the domination of clastic-organic and organic-clastic varves with lowest thicknesses and discontinuous basal detrital sublayers (Fig. 4 LZ IV) leading to generally higher counting uncertainties. Lowest deviations of <1 % in LZ II, LZ III, LZ IV and in LZ V represent best preservation and thus easily countable varves of different varve types. Generally, clastic-organic and clastic-calcitic varves with higher varve thickness, especially in LZ V are most reliably countable. The relatively high uncertainties of 7.7 % in LZ V are due to the low number of varves comprised in individual thin sections (Fig. 8). For the total uncertainty estimate for the floating varve chronology we use the mean of ± 5 % calculated from the uncertainties for each 10 cm interval. This conservative estimate is more realistic than the very low difference in the two repeated varve counts. An uncertainty of 5 % is in the range of varve chronologies reported elsewhere (Ojala et al., 2012).

Since the uppermost 63.0 cm of the sediment profile is largely homogeneous, the varve chronology is floating and needs to be anchored to an absolute chronology at this point. The interpolation with a mean SR derived from the combination of the consistent CIC and CRS [210]Pb marker (AD 1945), the SR derived from discontinuously varved sequences between 41.0 and 63.0 cm depth and from 100 measured varves below 63.0 cm depth seems to be the best approach for constraining the uppermost age-depth relationship within the homogenous sediments, where further chronological markers are lacking. We are aware, that the interpolation-based "floating" anchor point at 63.0 cm depth is prone to additional uncertainties. We considered this by assuming a higher uncertainty of 10 % for this interval, than that of ± 5 % for the floating varve chronology.

### 5.3 Radiocarbon reservoir effects

Compared to the floating varve chronology, including two terrestrial (wood) AMS [14]C dates, we observed a general trend of decreasing reservoir effects of dated aquatic material up core with the maximum deviation of ~6,150 years at 585.0 cm depth (10,930 ± 570 years BP) in LZ I (Fig. 7). The step-wise decrease of deviations between radiocarbon and varve ages is most pronounced at the boundary between lithozone LZ IV and LZ V, when it abruptly decreases from ~3,000 years to ~1,000 years. The reservoir effect generally depends on the rate of atmospheric $CO_2$ exchange between the water column and the air, internal mixing dynamics and the input of [14]C depleted carbonaceous material (Ascough et al., 2010; Jull et al., 2013; Keaveney and Reimer, 2012; Lockot et al., 2016; MacDonald et al., 1991). The catchment of Lake Chatyr Kol exhibits several sources that could be responsible for a [14]C-depletion of dissolved carbon species in the lake water. Highest reservoir ages in the early Holocene are likely the result of the combined influence of these sources: 1) the input of old, [14]C-depleted $CO_2$ with glacial meltwater (c.f. Hall and Henderson, 2001) at the onset of a warming Holocene and 2) the weathering and erosion of the northern

outcropping limestones, which led to the release and input of dissolved bicarbonate to the lake (c.f. Abbott and Stafford, 1996; Hutchinson et al., 2004). Both processes lead to a $^{14}$C-depleted $CO_2$ and $HCO_3^-$ uptake during photosynthesis by e.g. submerged aquatic plants like *Ruppia martima* at 585.0 cm depth (Fig.7) and by phytoplankton, on which daphnia feed and which therefore

also show similar reservoir effects. A high detrital input and thus a potentially high input of dissolved bicarbonate is supported by increased varve thicknesses during the early Holocene (Sect. 5.4.1, Fig. 9). Furthermore, 3) thawing of permafrost since the beginning of a warming Holocene might have released dissolved $^{14}$C-depleted organic material and thus affect the $^{14}$C TOC bulk measurements. Our fieldtrip observations and observations by Shnitnikov (1978) of modern permafrost reduction and the development of thermokarst in the southern part of the catchment around the neighbouring Lake Maloye (Fig. 1) support this

assumption. The cause of a step-wise reservoir effect reduction is therefore likely also related to the combined effect of a generally decreasing glacial influence and a decreasing input of bicarbonate until ~AD 1150 at the boundary between LZ IV and LZ V (Fig.7, 9). The abrupt decrease of the reservoir effect after ~AD 1150, despite an increase in detrital carbonate supply (Sect. 5.4.5, Fig. 9) might be related to the silting up of the basin leading to a shallower water depth, which is more susceptible to water circulation and an enhanced atmospheric $CO_2$ exchange (c.f. Geyh et al., 1997).

**5.4 Holocene variations in varve microfacies**

The Lake Chatyr Kol sediment profile comprises six different varve types (Sect. 4.2, Fig. 4.1, Suppl. Fig. 2), which occurrences showed varying dominances in the different lithozones that are described below. The individual lithozones always comprised more than one varve type (Fig. 9) with a maximum of five different varve types occurring in LZ III and II to three varve types in LZ V.

**5.4.1 Lithozone I (623.5-566.0 cm: 11,619 ± 603 to 10,730 ± 560 years BP)**

Lithozone I is characterized by relatively high and varying varve thicknesses and by the presence of clastic-organic, clastic-calcitic and clastic-aragonite varves (Fig. 9 LZ I). Clastic-organic varves constitute about 57 % of the observed and counted varve types, clastic-calcitic 29 % and clastic-aragonitic 14 %. Generally thick detrital coarse-grained spring sublayers, which

were observed in all varve types in this LZ, are indicative for intense runoff by winter/spring snow meltwater and/or by glacial thawing during summer (Shnitnikov, 1978) caused by highest insolation (Berger and Loutre, 1991; Chen et al., 2008; Jin et al., 2011; Li and Morrill, 2010) at the onset of a warming early Holocene. Glaciers of the inner Tian Shan started to retreat between ~12-8 ka years BP (Bondarev, 1997; Shnitnikov, 1978) causing enhanced detrital input into the lake. The low species abundances of aquatic plants (*Ruppia Maritima or Potamogeton* sp.), daphnia and characeae reflect a littoral community and

indicate a low aquatic productivity and a relative low lake level during this time. Clastic-calcitic varves appear at the base of the composite profile and towards the end of LZ I, whereas clastic-aragonitic varves dominate in the period from ~11,500 to 11,000 years BP. Most likely, idiomorphic aragonite formed due to a combination of Mg-rich water supply to the lake and strong evaporative conditions causing lake water Mg/Ca ratios of >12 (Kelts and Hsü, 1978; Müller et al., 1972).

**5.4.2 Lithozone II (566.0-480.0 cm: 10,730 ± 560 to 8,040 ± 430 years BP)**

In this lithozone calcitic-clastic varves constitute about 21 % of the observed varves and clastic-calcitic varves ~22 %, while clastic-organic varves make up ~42 % and clastic-diatom varves ~15% (Fig. 9 LZ II). This lithozone is characterized by intercalations of calcitic varve types (calcitic-clastic & clastic-calcitic) with clastic-organic and clastic-diatom varves (Fig. 4.2). The variations of these varve types might be related to external (climatic) forcing or lake-internal or sedimentation variability (Turner et al., 2016 and reference herein). XRF element maps show endogenic calcite sublayers that are enriched in Sr and Mg alternating with clastic (Si and Al) or diatom (Si) layers (Fig. 4.2) suggesting Sr- and Mg-rich calcite in this lithozone which indicates evaporative concentration (Müller et al., 1972). The shift of the dominant endogenic carbonate type from aragonite in LZ I to calcite in LZ II around ~10,730 years BP (Fig. 9 LZ II) coincides with an increase in biological and photosynthetic activity, as inferred from the establishment of a diverse lake fauna seen in high abundances of chrysophytes, planktic (*Cyclotella choctawhatcheeana)* and periphytic diatoms (*Achnanthes brevipes*) as well as of aquatic plants, ostracods, characeae and few daphnia. Identifying the main drivers controlling endogenic carbonate formation thus remains speculative. Species assemblages and associated biological activity during the summer season indicates favourable warm summers and sufficient nutrient supply through e.g. runoff. Because the species assemblages show mixed littoral and pelagic species abundances, these are interpreted as an indication for a low lake level.

**5.4.3 Lithozone III (480.0-273.0 cm: 8,040 ± 350 to 4,140 ± 230 years BP)**

The deposition of clastic-calcitic varves comprise ~38 % of the observed varves in this lithozone, while clastic-organic varves make up ~27 % and clastic-diatom varves ~34 % (Fig. 9 LZ III). Clastic-calcitic varves are generally thicker than the other varve types of this LZ mainly because of exceptionally thick summer sublayers. These summer layers consist of endogenic calcite mixed with resuspended calcites and fine-grained detrital grains (Fig. 4.1c, Fig. 4.3 a). This is confirmed by elevated Sr values of the XRF element mapping, indicating the presence of Sr-rich carbonates. These varves likely reflect increased resuspension of carbonates from the littoral zone due to wind induced wave activity. As in LZ II, alternations of clastic-calcitic (Ca, Sr) (Fig. 4.3 a), clastic-diatom and clastic-organic (Si, Al) (Fig. 4.3 b) varves are characteristic for this lithozone as well. At ~8,040 years BP the deposition of calcitic-clastic varves ceased and is replaced by the deposition of clastic-calcitic, clastic-organic and clastic-diatom varves probably due to decreasing summer insolation and lower summer temperatures (Berger and Loutre, 1991; Chen et al., 2008; Jin et al., 2011; Li and Morrill, 2010). In addition, higher lake levels since that time are indicated by the dominance of planktic diatoms and the occurrence of lake deposits at the eastern and southern shore which have been dated from 6,688 ± 473 to 4,621 ± 594 cal years BP ([14]C ages published by Shnitnikov (1978) calibrated with OxCal4.3 & IntCal13). During our field work we found lake sediments on a shallow terrace ~7 m above the current lake level east of the lake which also revealed a mid-Holocene age of 5,786 ± 122 cal years BP (Poz-109830 Tab.2). Higher lake levels at that time have been explained by the preceding early Holocene glacier retreat in the catchment (Bondarev, 1997; Shnitnikov,

1978). However, more recently even minor glacial advances in the Aksai Basin east of the Chatyr Kol catchment [10]Be exposure dated between 7.5 and ~4.5 ka were reported (Koppes et al., 2008).

**5.4.4 Lithozone IV (273.0-130.0 cm: 4,140 ± 230 to 800 ± 60 years BP/ AD 1150 ± 60**

This Lithozone contains clastic-organic (58 %), organic-clastic (18 %), clastic-diatom (17 %) and clastic-calcitic varves (6 %) (Fig. 9 LZ IV). High abundances of planktic diatoms as well as clastic-diatom varves prevail until ~2,900 years BP, whereas clastic-organic and organic-clastic varves with abundant aquatic plant remains and periphytic diatoms occur afterwards and dominate the sediments particularly after ~2,200 years BP. Abundant aquatic plant remains and periphytic diatoms can be explained by reworking due to wave activity and water column mixing during low lake levels. Low lake levels are inferred from the modern observation that aquatic plants occupy the shallow parts of the Lake from ~15.0 to 0.5 m. A decreased lake level combined with the shallow bathymetry of the lake basin (Fig. 1) further promotes large impacts on the species communities, which supports the changing abundances from a planktic to a littoral dominated fauna and flora after ~2,200 years BP. An increased mixing of the water body also caused a clear decline in varve preservation. The rate of detrital input as observed in clastic-organic, organic-clastic varves (Fig. 4.1d, Suppl. Fig. 2e) is rather constant and mainly appears within the spring sublayer suggesting stable snow melt runoff from 4,140 years BP to AD 1150. At ~2,600 and at 2,300 years BP clastic-calcite varves with thickened summer sublayers appear for a few decades indicating enhanced resuspension.

**5.4.5 Lithozone V (130.0-41.0 cm: AD 1150 ± 60 to AD 1730 ± 30)**

Clastic-organic varves constitute 59 % of the observed varves in LZ V, clastic-calcitic varves 26 % and organic-clastic varves 15 % (Fig. 9 LZ V), the latter ceasing at 110.5 cm (AD 1260 ± 50). Varve microfacies changes abruptly at 130 cm depth or AD 1150 from the dominance of organic-clastic varves to dominating clastic-organic and clastic-calcitic varves. Within 5 years, varve thickness drastically increased from Ø 0.43 mm in LZ IV to Ø 1.52 mm in LZ V due to thicker summer sublayers. Thicker summer sublayers result from thicker mixed sublayers rich in algal remains (*Botryococcus,* chrysophytes, diatoms) and additional late summer detrital sublayers (Fig. 4.1.e, Suppl. Fig. 2f). Hence, the increase in summer layer thickness suggests both higher lacustrine productivity and an increase in summer runoff events. However, the reasons for these changes remain elusive and a relationship to known climatic periods like the Medieval Climate Anomaly and the Little Ice Age is not found. One might speculate that the frequent occurrence of late summer runoff layers either reflects convective rainfall events due to recycling of local moisture sources (Aizen et al., 2001) or changing atmospheric circulation regimes. Changes in boundary conditions in the catchment of the lake are unlikely since microfacies analysis does not show pronounced changes in grain size distribution of the detrital material. Human impact cannot fully be excluded but low indices of human and livestock fecal biomarkers (Schroeter et al., 2020) are an argument against major human impact. The presence of lake deposits at the northern and southern shores ca 1.5 - 1 m above present day lake level dated at AD 1420 ± 204, AD 1044 ± 160 and AD 858± 166 (Shnitnikov, 1978) suggests that increased summer runoff events might have resulted in a more positive water budget and lake level rise.

### 5.4.6 Lithozone VI (41.0-0.0 cm: AD 1730 ± 30 to AD 2012)

At around AD 1730 ± 30 varve formation and/or preservation ceased and sediments became predominantly homogeneous. The cessation of varves might be related to enhanced mixing of the water column resulting in a loss of the oxygen minimum zone (Fig. 2a) caused by decreasing water depth due to silting–up, which accelerated with the abrupt increase in sedimentation rate at AD 1150 and/or due to strengthening of the wind conditions and wave activity.

### 6. Conclusion

We present the first varved lake sediment record in arid Central Asia that covers almost the entire Holocene. The established floating varve chronology provides independent dating for a setting with scarce material for radiocarbon dating. In particular, our varve chronology allows a quantification of changes in radiocarbon reservoir ages throughout the Holocene. The largest reservoir effect of ~6150 years in the early Holocene is likely caused by glacial melt and enhanced local erosion resulting in a surplus of dead carbon. Lowest reservoir ages of ~1,000 years and less in the late Holocene might be related to enhanced atmospheric $CO_2$ exchange when the lake was shallower due to silting-up of the lake basin and/or increased windiness inducing increased water column mixing and $CO_2$ exchange with the atmosphere. The construction of the varve-based chronology was only possible through detailed micro-facies analyses of the entire sediment sequence in overlapping thin sections that allowed the development of seasonal deposition models for all observed types of fine laminations. Based on these models and their comparison with published varve micro-facies data, we interpret all six Chatyr Kol lamination types as varves. Compared to many other varved lake sediment records, the Chatyr Kol varves are very heterogeneous and a complex pattern of six different micro-facies types developed throughout the Holocene. All varve types are predominantly clastic and comprise variations of their summer sublayers with changing dominances of organic, diatom, calcitic, aragonitic and additional detrital sublayers. Varve thickness changed accordingly with the varve micro-facies types, whereby the most conspicuous increase of varve thickness occurred at AD 1150 which is caused by increased erosion and runoff. The increase in detrital input into the lake further caused an acceleration of the silting-up processes.

XRF element mapping results support our microfacies analysis and provide additional information on the composition of carbonate sublayers and detrital carbonate. Microfacies analysis and XRF element mapping show major variations between partly Mg and Sr rich sublayers in calcitic-clastic and clastic-calcitic varves with Al and Si rich sediments in clastic-organic and clastic diatom varves. Nevertheless, the complex succession and variations of varve types throughout the Holocene including major change points still requires further detailed investigations and interpretation together with other proxy data.

### Data availability

The presented data is provided through PANGAEA: https://doi.pangaea.de/10.1594/PANGAEA.909981

and at https://varve.gfz-potsdam.de.

## Competing interest

The authors declare that they have no conflict of interest.

## Authors contributions

JK performed the microfacies analysis, $^{210}$Pb and $^{137}$Cs gamma spectrometry and wrote the manuscript with contributions from all co-authors. JM designed the project and organised field work. SL carried out sediment coring and was responsible for $^{14}$C dating. RU provided information about the catchment geology and lake level changes. RT carried out the XRF element mapping. AB supervised analyses and manuscript writing.

## Acknowledgements

We are grateful for the suggestions of three anonymous reviewers and R. Staff, which helped to improve the manuscript. We thank S. Lauterbach, S. Pinkerneil, M. Köhler, R. Schedel, and D. Henning for retrieving the long piston cores from Lake Chatyr Kol in 2012 in the framework of the project CADY (Central Asian Climate Dynamics, BMBF grant 03G0813). S. Pinkerneil and Y. Beutlich are thanked for help with the geochemical analyses. S. Orunbaev, M. Daiyrov, S. Kalmuratov, G. Omurova and K, Jusupova are acknowledged for their support during field trips. We further thank T Goslar for AMS $^{14}$C
dating and D. Berger, G. Arnold and B. Brademann for thin section preparation and G. Schettler for his help with lead-210 dating. We also want to thank Rik Tjallingii for conducting µXRF element mapping and his help with the revision of the manuscript. This paper is a contribution to Topic 8 'Rapid Climate Change from Proxy data' within the climate initiative REKLIM of the Helmholtz-Association.

## Funding

This study was conducted in the framework of CAHOL (Central Asian HOLocene), a subproject of the joint project CAME II (Central Asia Climate Tipping Points and their Consequences) funded by the German Federal Ministry of Education (BMBF) through grant 03G0864B.

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

**Tables**

| Core ID | Latitude | Longitude | Depth (m) |
|---------|----------|-----------|-----------|
| CHAT12 | 40°36.370 | 75°14.020 | 20 |
| SC17_1 | 40°36´756 | 75°14´481 | 15.05 |
| SC17_2 | 40°36´587 | 75°15´138 | 17.25 |
| SC17_3 | 40°36´315 | 75°14´577 | 18.25 |
| SC17_4 | 40°39´124 | 75°19´891 | 5-10 |
| SC17_5 | 40°36´363 | 75°14´079 | 19.5-20 |
| SC17_6 | 40°36´213 | 75°13´939 | 19.2 |
| SC17_7 | 40°36´147 | 75°14´062 | 18.5 |

**Table 1: Coordinates of long and short cores.**






| Depth (cm) | LabID | $^{14}$C | Error | cal. years BP (95.4% probability range) | cal a BP (midpoint + span) | material |
|---|---|---|---|---|---|---|
| 0 | Poz-109830 | 5050 | 40 | 5664-5908 | 5786±122 | leaves |
| 0 | Poz-54280 | 330 | 30 | 308-473 | 391±83 | recent aquatic plant |
| 0 | Poz-54281 | 2425 | 25 | 2354 - 2692 | 2523±169 | recent aquatic plant |
| 0 | Poz-54279 | 225 | 30 | -4 - 310 | 155±155 | recent *Daphnia* |
| 38.5 | Poz-54282 | 755 | 35 | 660 - 735 | 697±36 | aquatic plant |
| 41.5 | Poz-56609 | 1265 | 30 | 1088 - 1285 | 1187±98 | bulk TOC |
| 59.5 | Poz-54283 | 955 | 30 | 798 - 964 | 881±83 | aquatic plant |
| 88 | Poz-54286 | 1595 | 30 | 1410 - 1549 | 1480±69 | aquatic plant |
| 98 | Poz-54284 | 1715 | 35 | 1552 - 1706 | 1629±77 | aquatic plant |
| 98.7 | Poz-56614 | 1730 | 30 | 1564 - 1708 | 1636±72 | aquatic plant |
| 98.7 | Poz-556592 | 2220 | 30 | 2152 - 2324 | 2238±86 | bulk TOC |
| 111 | Poz-54287 | 1925 | 30 | 1817 - 1947 | 1882±65 | *Daphnia* remain |
| 115.5 | Poz-54288 | 1960 | 30 | 1830 - 1989 | 1910±79 | *Daphnia* remain |
| 179.5 | Poz-56596 | 4150 | 35 | 4572 - 4827 | 4700±128 | bulk TOC |
| 209.7 | Poz-56610 | 4930 | 35 | 5597 - 5727 | 5662±65 | bulk TOC |
| 229.5 | Poz-54289 | 5790 | 50 | 6415 - 6665 | 6540±125 | *Daphnia* remain |
| 252 | Poz-56595 | 5840 | 40 | 6533 - 6747 | 6640±107 | bulk TOC |
| 255.7 | Poz-54290 | 5880 | 35 | 6637 - 6785 | 6659±126 | *Daphnia* remain |
| 299.7 | Poz-56593 | 6840 | 40 | 7591 - 7757 | 7674±83 | bulk TOC |
| 345 | Poz-56594 | 7610 | 40 | 8025 - 8180 | 8103±78 | bulk TOC |
| 345 | Poz-54292 | 7305 | 35 | 8350 - 8514 | 8432±82 | bulk TOC |
| 370.1 | Poz-56613 | 8200 | 50 | 9015 - 9300 | 9158±143 | bulk TOC |
| **380.5** | **Poz-63307** | **5360** | **40** | **6003 - 6277** | **6140±137** | **wood** |
| 391.4 | Poz-54294 | 8550 | 50 | 9465 - 9604 | 9535±70 | *Daphnia* remain |
| 391.7 | Poz-54293 | 8710 | 50 | 9546 - 9887 | 9717±171 | *Daphnia* remain |
| 437.2 | Poz-54296 | 9160 | 50 | 10230 -10487 | 10359±129 | *Daphnia* remain |
| 439.9 | Poz-56611 | 9360 | 50 | 10427 - 10713 | 10570±143 | bulk TOC |
| 466 | Poz-54297 | 9670 | 50 | 10789 - 11211 | 11000±212 | *Daphnia* remain |
| 469 | Poz-54298 | 9690 | 50 | 10795 - 11226 | 11011±216 | *Daphnia* remain |
| 508 | Poz-54299 | 10840 | 50 | 12681 - 12804 | 12743±62 | *Ruppia maritima* |
| 510 | Poz-54300 | 11060 | 50 | 12790 - 13062 | 12926±136 | bulk TOC |
| 528 | Poz-54301 | 12150 | 50 | 13831 - 14175 | 14003±172 | bulk TOC |
| **528** | **Poz-54302** | **8890** | **50** | **9785 - 10191** | **9988±203** | **deciduous wood** |
| 549.5 | Poz-56591 | 12820 | 60 | 15105 - 15550 | 15328±223 | bulk TOC |
| 571 | Poz-56608 | 13220 | 70 | 15660 - 16125 | 15893±233 | bulk TOC |
| 585 | Poz-63308 | 14060 | 90 | 16759 - 17419 | 17089±330 | *Ruppia maritima* |
| 620.5 | Poz56590 | 13190 | 70 | 15612- 16092 | 15852±240 | bulk TOC |

**Table 2: $^{14}$C dates (calibrated with OxCal 4.3, IntCal13).**



**Figures**

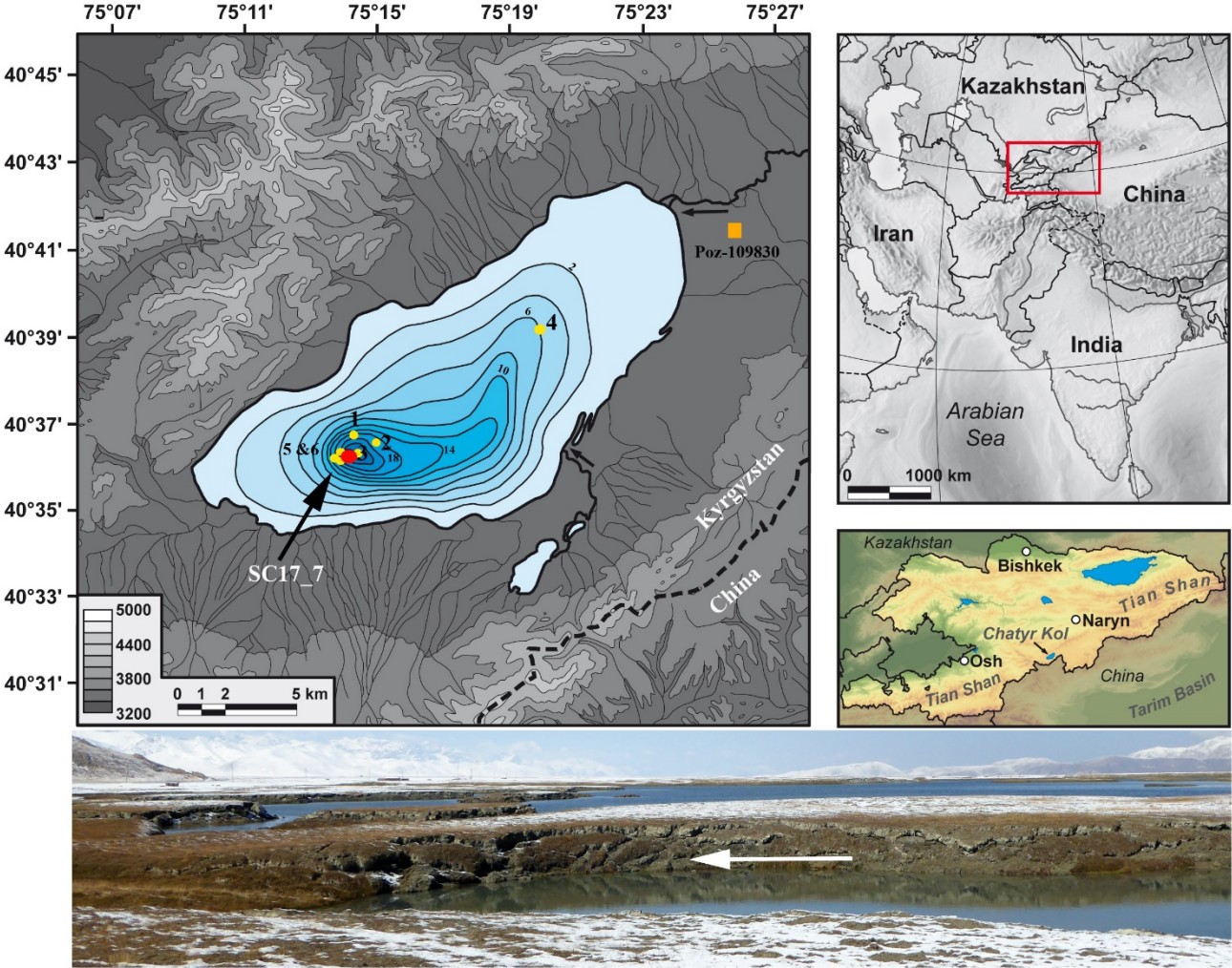

Figure 1: Location of Lake Chatyr Kul, the composite profile (red dot) and the gravity cores (yellow dots). The orange square marks the location of [14]C dated leaves (Poz-109830, Tab. 2) found in the top of a mid-Holocene-shoreline at ~3540 m a.s.l. The relief map of Kyrgyzstan relies on the CGIAR-CSI SRTM 90m (3 arcsec) digital elevation data (Version 4) of the NASA Shuttle Radar Topography Mission (Jarvis, 2008). The figure was modified from Lauterbach et al. (2014). Photo of unstable shores (white arrow) of Maloye lake.

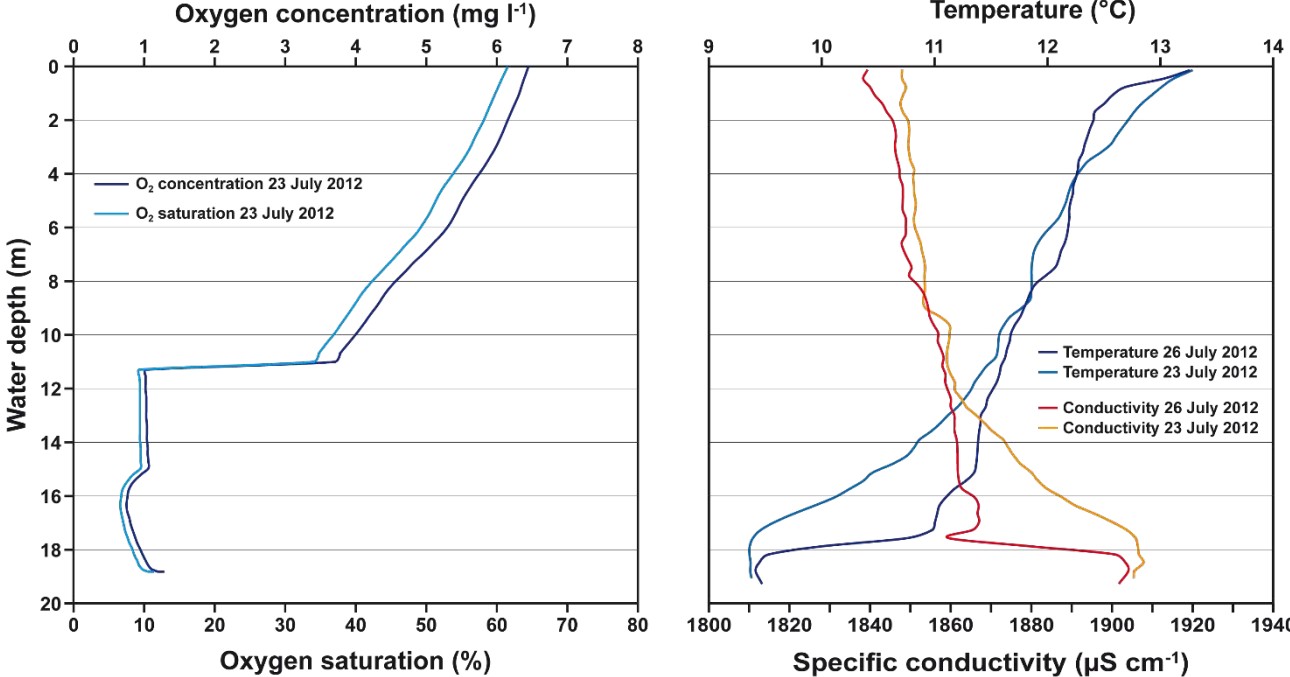


**Figure 2: Oxygen concentration (YSI Pro 6600 V2), Temperature and specific conductivity measured with a CTD sensor during the field trip in 2012 at the core´s location (N 40°36.371´, E 75°14.006´).**




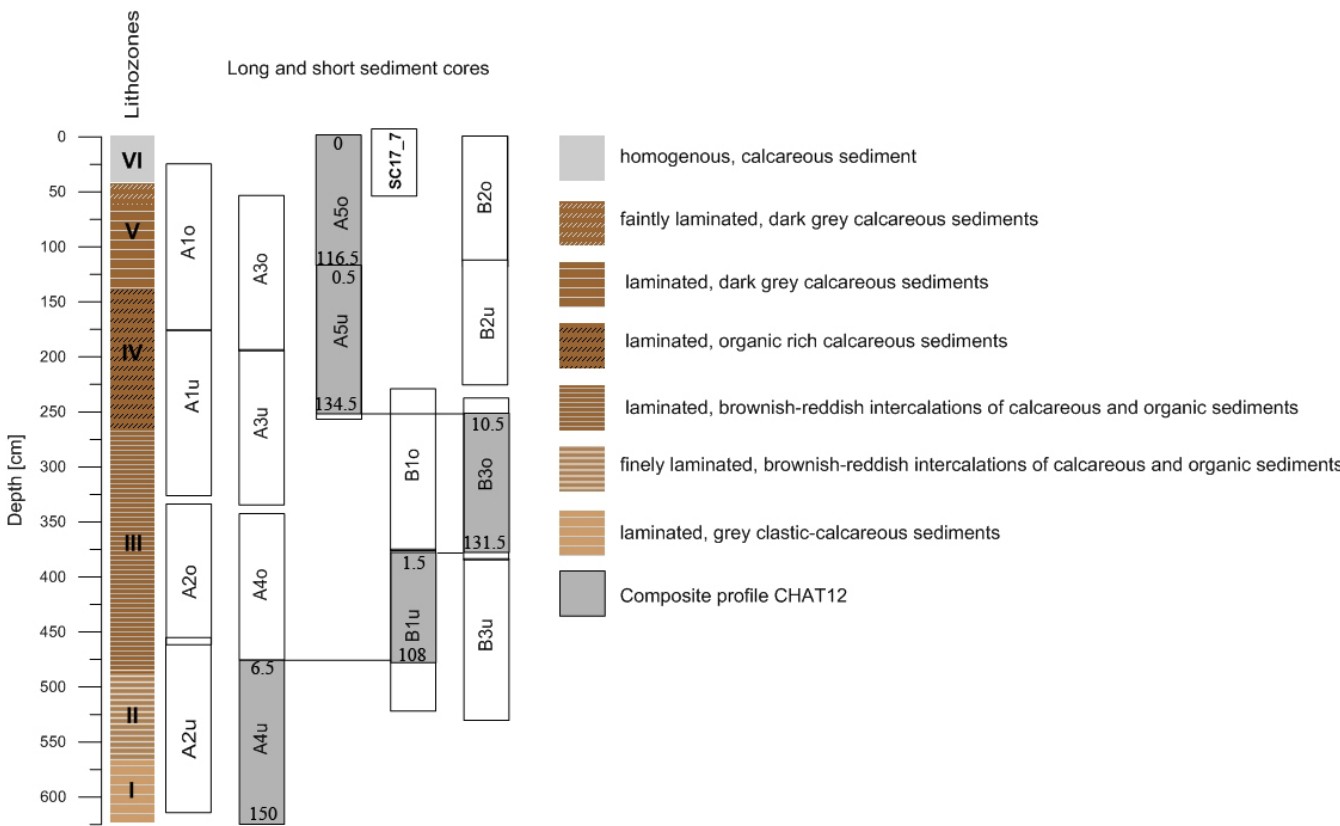

Figure 3: The composite profile CHAT12 (dark grey: piston cores A5o, A5u, B3o, B1u and A4u, used depth sections are displayed within). Additional gravity cores taken in 2017 (SC17_1 to SC17_7) were only partly used for thin section preparation and gamma spectrometry dating. The gravity cores cover approximately the upper first meter of the composite profile.


varve deposition models

thin section pictures of dominant varve types in the different lithozones

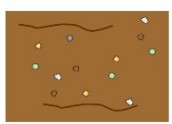

**homogenous sediments**

with faint laminae in the upper 1.5 cm depth (grey bar)

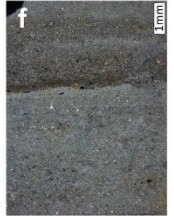 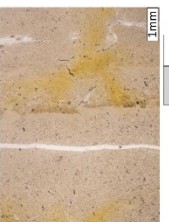

**clastic-organic laminations**

**Winter** amorphous organic layer
**Summer** runoff event (clastic varves in LZ V)
**Summer** mixed layer containing fine detrital grains and endogenic calcite
**Spring** coarse detrital runoff layer often with pennate diatoms and chrysophytes

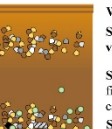 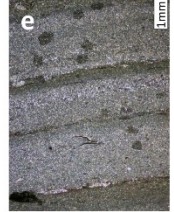 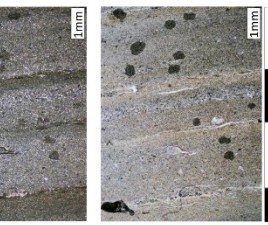

**organic-clastic laminations**

**Winter** amorphous organic layer
**Summer** mixed layer containing detrital grains, rich in aquatic plant remains & periphytic diatom *Achnanthes brevipes*, daphnia, ostracods, characeae
**Spring** coarse detrital runoff layer often with pennate diatoms and chrysophytes

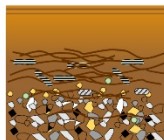 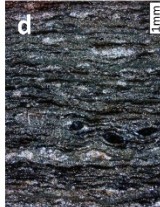 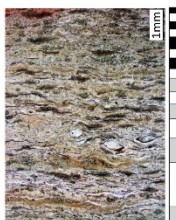

**clastic-calcitic laminations**

**Winter** amorphous organic layer

**Summer** mixed layer of endogenic calcite formation and fine detrital grains

**Spring** coarse detrital runoff layer often with pennate diatoms and chrysophytes

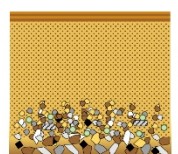 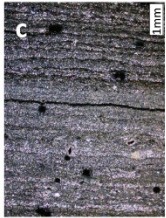 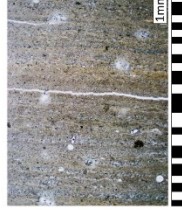

**clastic-diatom laminations**

**Summer/Autumn** *Cyclotella choctawhatcheeana* blooms
**Summer** mixed layer
**Spring** coarse detrital runoff layer often with pennate diatoms and chrysophytes

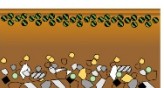

**calcitic-clastic laminations**

**Winter** amorphous organic layer

**Summer** layer of intensive endogenic calcite formation with scattered detrital grains

**Spring** coarse detrital runoff layer

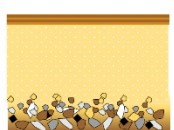 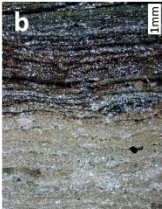 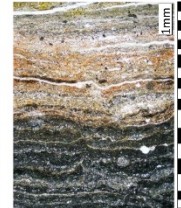

**clastic-aragonitic laminations**

**Summer** mixed layer of fine detritus and endogenic calcite with occasionally aragonite formation

**Spring** coarse detrital runoff layer rich in pyrite

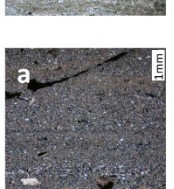 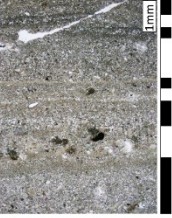
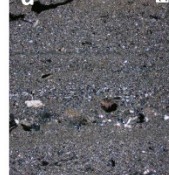 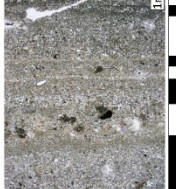
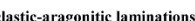

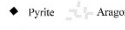 Pyrite 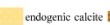 Aragonite 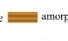 endogenic calcite 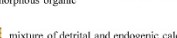 amorphous organic

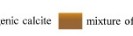 Coarse and fine detrital grains (qz, fsp, ca, mica) 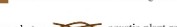 mixture of detrital and endogenic calcite mixture of fine detritus and organics

planktic diatom *Cyclotella choctawhatcheeana* periphytic diatom *Achnanthes brevipes* chrysophytes aquatic plant remains

**Figure 4:1** Thin section pictures of different lamination (varve) types in cross-polarized (left) and plane polarized (right) light of the different lithozones LZ I to LZ V. **a)** Clastic-organic laminations; **b)** Intercalation of clastic-organic, clastic-diatom and calcitic-clastic laminations; **c)** Intercalation of clastic-calcitic, clastic-organic and clastic-diatom laminations (upper part); **d)** Organic-clastic laminations; **e)** Clastic-organic laminations; **f) homogenous sediments.** Black/white/grey bars alongside the thin section pictures indicate one individual varve, but the reliability of grey bars (varves) is being generally lower due to high amounts of aquatic plant remains and low preservation. Process-related deposition Models of the observed lamination/varve types illustrate the seasonal depositional successions (a-e).

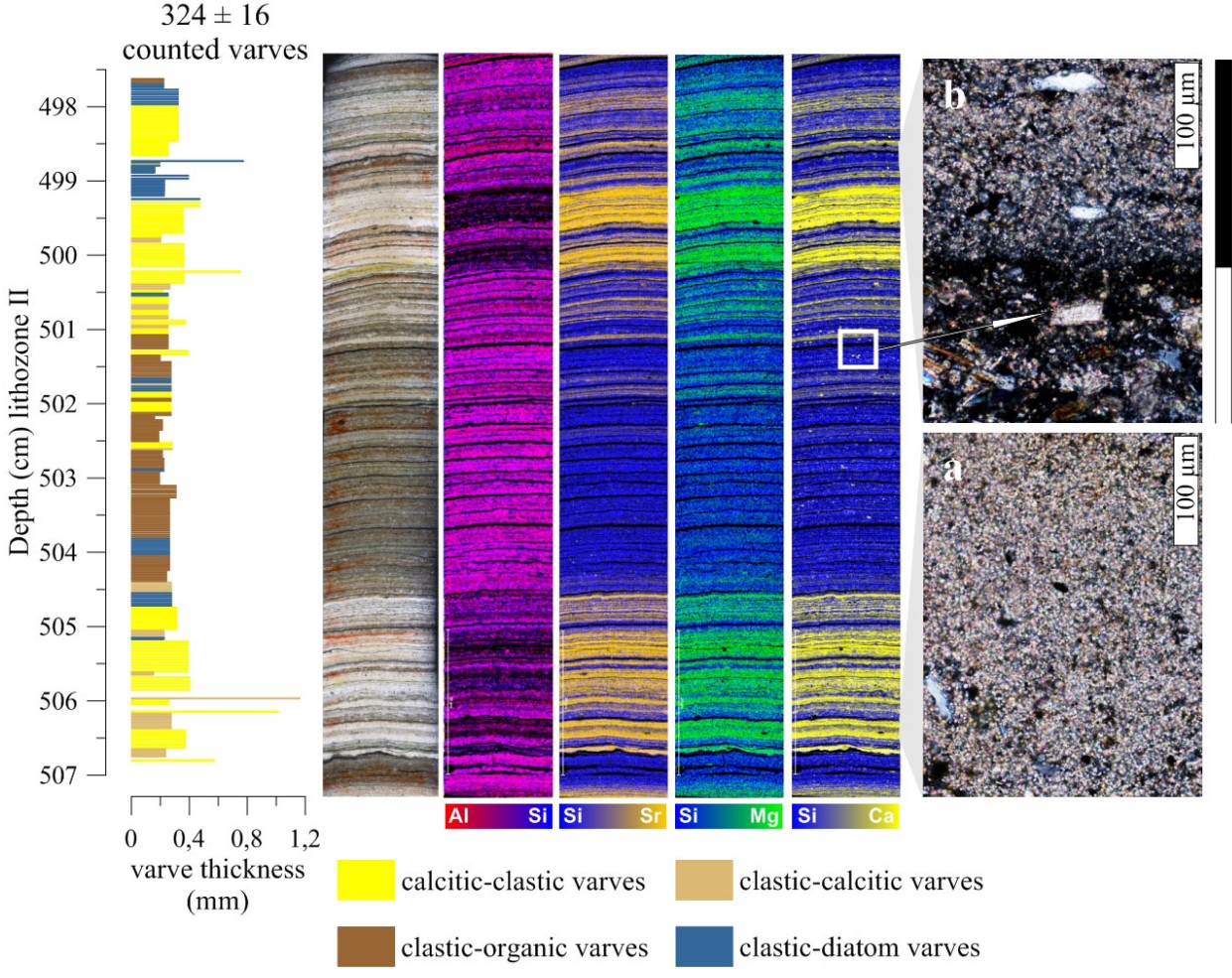

**Fig. 4.2 XRF-map 1: Varve types and XRF element mapping of a thin section from LZII from 507 to 497.5 cm depth. Element maps show an alternation of siliciclastic sediments with high amounts of Si and Al with calcite layers (Ca) with high amounts of Sr and Mg according to the presence of clastic-organic, clastic-diatom and clastic-calcitic and calcitic-clastic varves respectively. Microfacies analyses show endogenic calcite within calcitic-clastic varves within the summer sublayer (image a). Clastic-organic and clastic-diatom varves are indicated by high amounts of Si and Al with only individual calcite (arrow image b) and Mg-calcite grains. Note, that coinciding occurrences of individual elements results in colour mixing e.g. Al (red) and Si (blue) becomes pink. Endmembers of the Al-Si map are indicative for the presence of Al-rich clays and diatomaceous Si.**



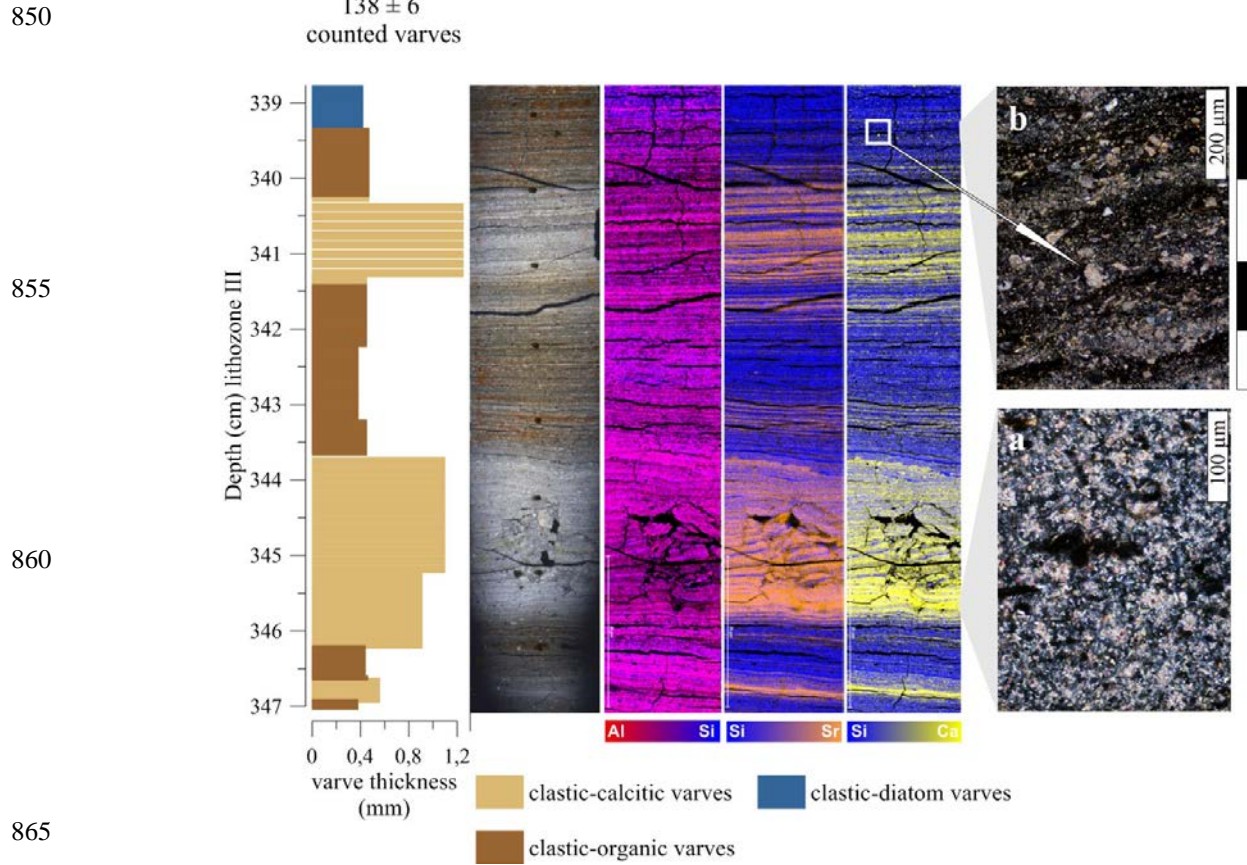

**Fig. 4.3 XRF-map 2: Varve types and XRF element mapping of a thin section from LZIII from 357 to 338.5 cm depth. Element maps show an alternation of siliciclastic sediments with high amounts of Si and Al with calcitic layers (Ca) with high amounts of Sr according to the presence of clastic-organic, clastic-diatom and clastic-calcitic varves respectively. Micro-facies analyses show mixed calcitic (resuspended and endogenic) summer sublayers within clastic-calcitic varves (image a). Clastic-organic and clastic-diatom varves are indicated by high amounts of Si and Al with only individual calcite grains (arrow image b). Note, that coinciding occurrences of individual elements results in colour mixing e.g. Al (red) and Si (blue) becomes pink. Endmembers of the Al-Si map are indicative for the presence of Al-rich clays and diatomaceous Si (blue).**

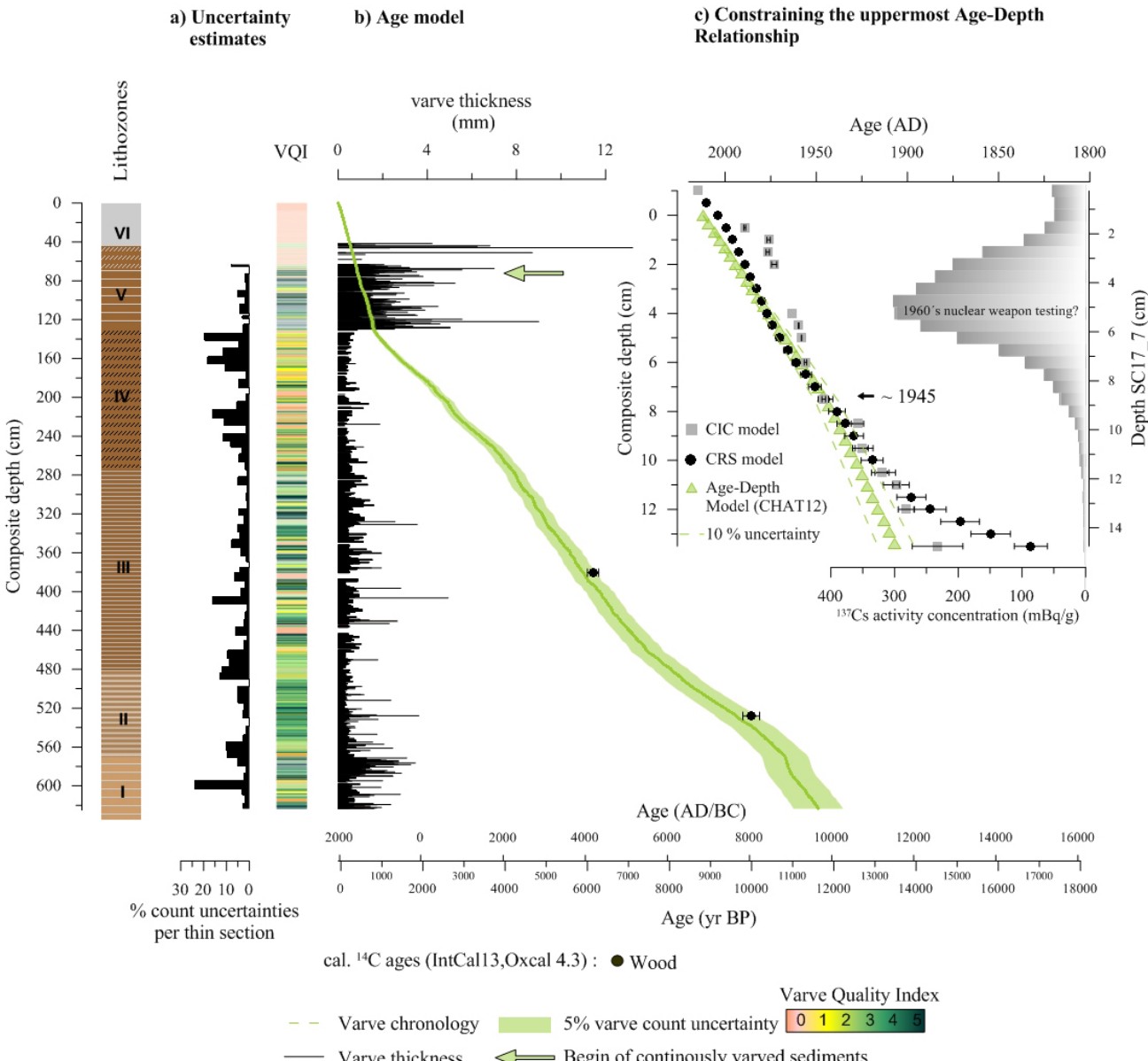

**Figure 5: a) Varve counting uncertainty estimates (mean of 5% =green) and VQI distribution. b) Age model of the floating varve chronology (Chatvd19) from 63.0-623.5 cm depth with a basal age of 11619 ± 603 years BP (light green). The black line shows the measured varve thickness, black dots mark the distribution of calibrated AMS 14C ages (with 95.4 % probability range) of wood pieces (Tab.2). b) ²¹⁰Pb CIC (grey squares) and CRS (black dots) age models, ¹³⁷Cs activity concentration profile and constrained age model for the uppermost part of the composite profile (green triangles).**

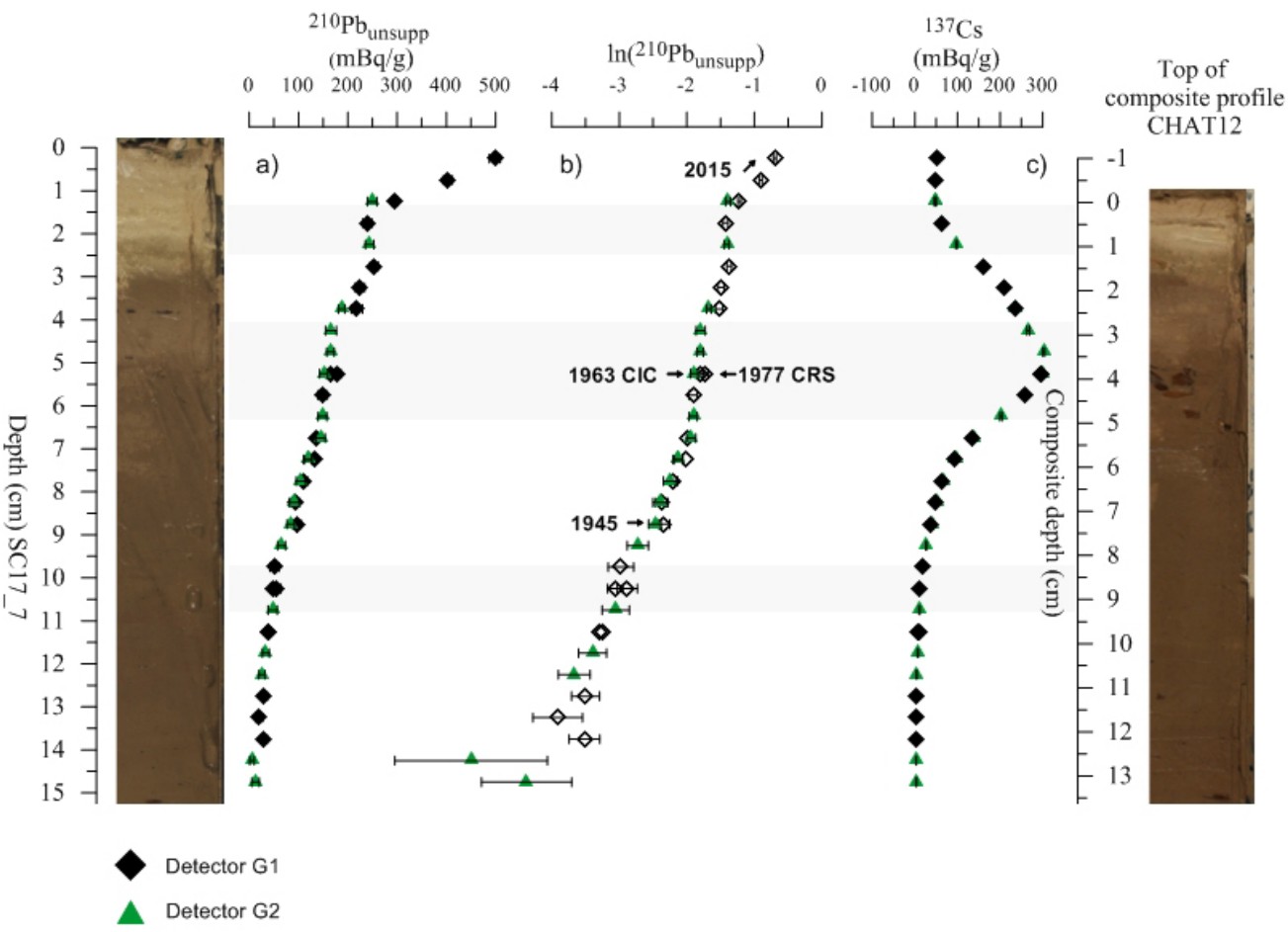

**Figure 6: Gamma spectrometry results of the gravity core SC17_7. Light grey intervals indicate uncorrelated sequences of the ln$^{210}$Pb$_{unsupp}$ vs. depth profile which affected the CIC model calculations (Suppl. Fig.2, Suppl. Tab. 2, 3). Core pictures of the upper part of the composite profile CHAT12 (right) and the gravity core SC17_7 (left) illustrate the facies change to calcite-enriched sediments in the uppermost centimeter.**

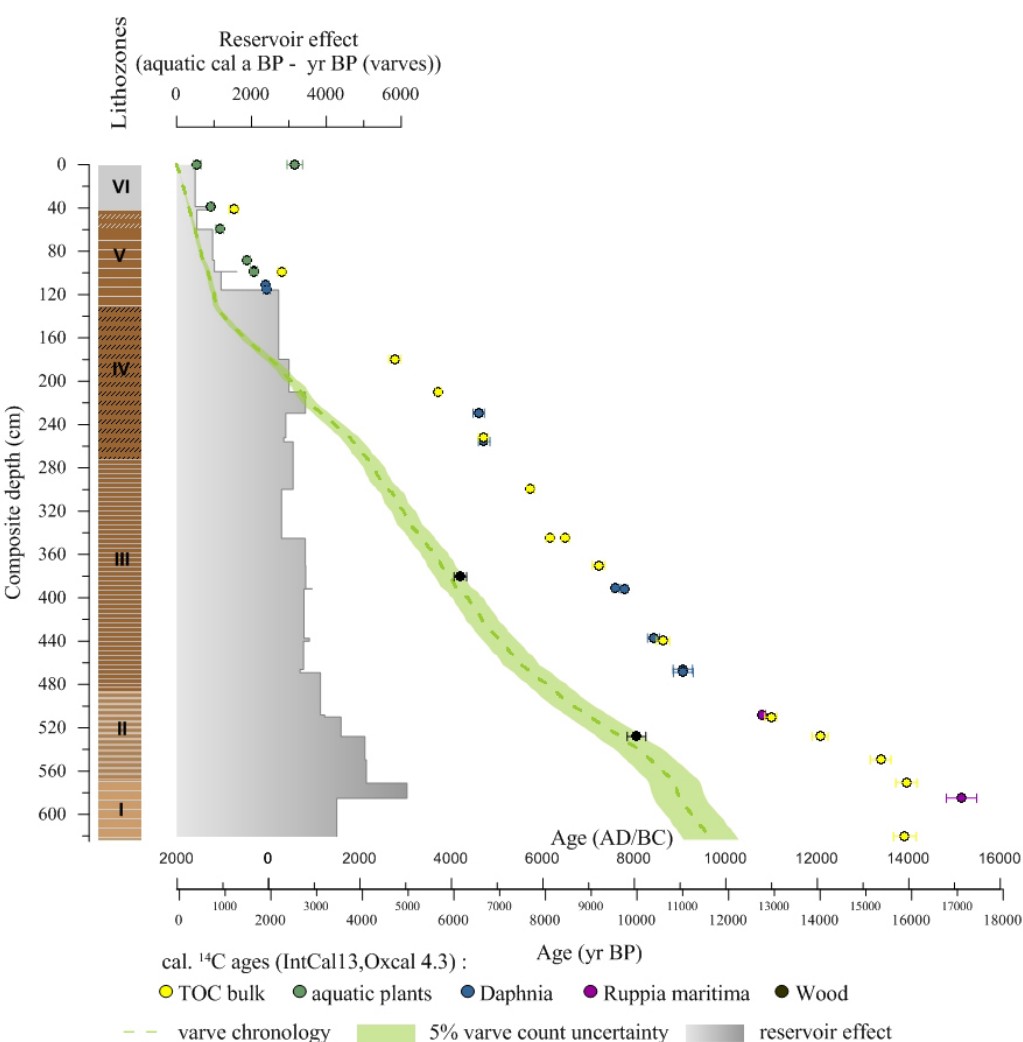

**Figure 7:** Radiocarbon reservoir effect (grey step plot). The reservoir effect was determined by the difference of aquatic cal. a BP (colored symbols) and varve ages (green). Floating varve chronology (green) and distribution of calibrated AMS $^{14}$C ages (with 95.4 % probability range) of wood pieces, TOC bulk, aquatic plant remains, daphnia and *Ruppia maritima* remains (Tab.2).

920

925

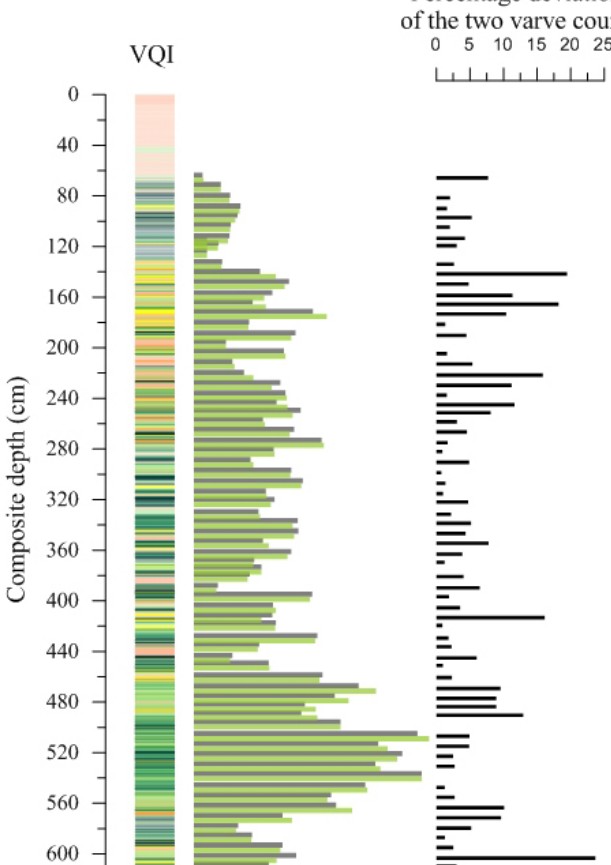

**Figure 8: VQI, Counting differences for individual thin sections (green= 1st count, black = 2nd count, and their percentage deviation.**

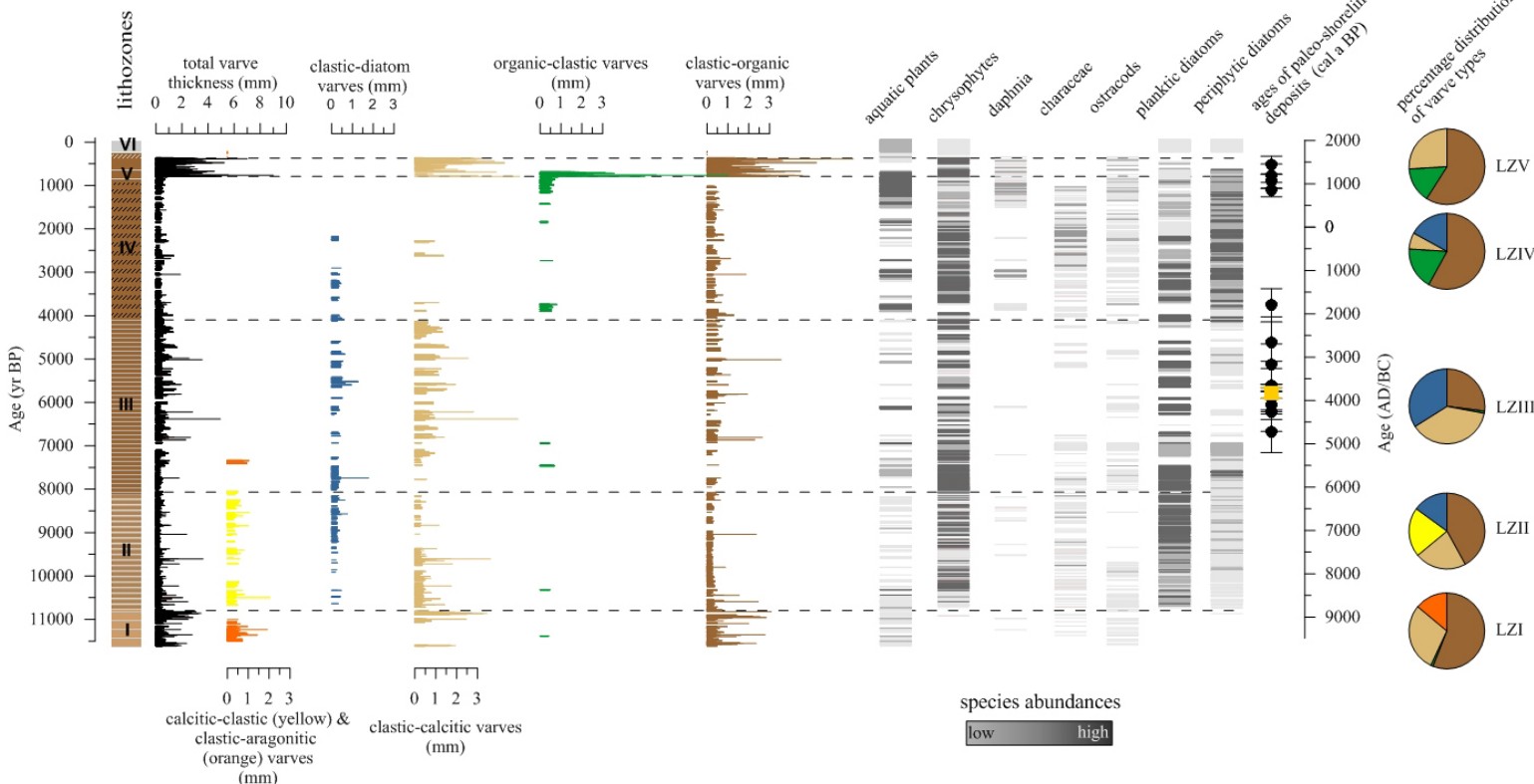

**Figure 9: Holocene seasonal deposition patterns, semi-quantitative species assemblages and calibrated [14]C dates of paleo-shore deposits (black dots from Shnitnikov (1978), yellow square from own sample taken in 2017 (Table 2). Percentage distribution of varve types in the different lithozones.**