# Peer review of "Seasonal deposition processes and chronology of a varved Holocene lake sediment record from Lake Chatyr Kol (Kyrgyz Republic)"

_Geochronology, 2019_

## Referee Comment (RC1) · Anonymous Referee #1 · 9 Feb 2020

Manuscript number: gchron-2019-18 Title: Seasonal deposition processes and chronology of a varved Holocene lake sediment record from Lake Chatyr Kol (Kyrgyz Republic)

This manuscript of Kalanke et al. is a very well written, detailed and thoroughly discussed manuscript focused on the geochronology of the sedimentary record of a small lake in the Kyrgyz Republic. The focus on dating of the sediments, the discussion of varve structure and composition, and the combination with radiocarbon, lead-210 and cesium-137 dating makes this manuscript ideally suited for this journal. Furthermore, the authors also discuss the sediment fine structure and try to relate the sediment mi-

crofacies to paleoclimate changes. I enjoyed reading this manuscript, the figures are excellent, and all figures and tables are easy to understand. The size of the article seems appropriate (or slightly shortened) and I have only few comments, so I would recommend considering publication of this manuscript after inclusion of the minor suggestions below.

The only major suggestion would be to include more information and interpretations of the data regarding environmental and climate changes that the lake has experienced during deposition of the sedimentary record. In Lines 54-55 it is specifically stated that the main aim of the projects is a better understanding of Holocene climate in Central Asia, so this should also be included in this paper. It is the only main weakness of the manuscript and the paper would have much more value if more details on the climate changes in the region are included as well. That is why a third focus should be included in line 57, which states something like "3) reconstruction of regional climate changes in Central Asia". The discussion of climate changes could partly be included into chapter 5.1 or within the last chapter of the discussion, to extend what the authors describe very briefly. This could either be added to chapter 5.4 or as a new chapter 5.5.

Specific, minor comments:

Lines 14-17. These first two sentences of the abstract appear to be repetitive and could be combined in shortened form.

Line 44: Typo – "n" to be removed

Lines 53-54: This information should be included into supplements. The aim of these projects is obviously the reconstruction of Holocene climate and so more information on this should be provided in the paper.

Line 62: remove dash

Line 70: delete amounts

Line 133: How long was the in-growth time (check also spelling to change to "in-growth

time"). The term photo peak activity sounds incorrect to me and should be replaced with a more appropriate term. It is the gamma energy that is recorded in the gamma spectrometry.

Line 139: lab-internal – please note which lab, and where these samples were analysed.

Line 167: use a,b,c to refer to each group of laminae more easily and use this instead of "LZ+number" in the references to Fig 4 throughout the text.

Line 204: Add current institute/university of Ms Schwarz within brackets as well.

Line 231: Add picture of homogeneous sediment to Fig 4 as well to see how it compares to the varved intervals. In particular, this is useful to show the faint, discontinuous laminae in the uppermost cm.

Lines 269-270: Is this assumption justified? +/- 40 years BP uncertainty could be higher or lower? Why is it not possible to more precise that this? On the basis of the data presented, I would be surprised if the error is as high as 40 years?

Line 386: change to effect

Line 491 (data availability statement): Please add the data into this database during the review process, so you can include the doi of the dataset in this statement. I think it is very important to add the doi to the final paper, so the future reader can access the datasets easily.
* * *

---

## Referee Comment (RC2) · Anonymous Referee #2 · 19 Feb 2020

The paper entitled "Seasonal deposition processes and chronology of a varved Holocene lake sediment record from Lake Chatyr Kol (Kyrgyz Republic)" from Kalanke et al. presents a very detailed and almost continuous varved record spanning the past ~11,000 years in a region where high-resolution paleoclimate data is currently lacking. This makes this record very exciting for the overall paleoclimate community. The MS is very well written, and the figures are excellent and easy to understand. Many dating techniques are shown including Pb-210 and Cs-137, radiocarbon and most importantly, varve counting. Hence, this paper is clearly suited for this journal.

[Figure]

Although I have no major comment on the central topic of this paper (that is suitable for the journal), i.e. the chronology, I am puzzled why there is no $\mu$-XRF data (e.g. Itrax) shown in your study. For example, the authors describe periods of prevailing anoxic bottom water conditions, calcitic materials/diatoms, coarse vs finer sediments, etc. In my opinion, it would be very helpful to show $\mu$-XRF elements (and elemental ratios) to support your visual microscopic analysis. Have you made such analysis (XRF)? If you are to interpret the paleoenvironments from this site in the paper, I think that would be very valuable.

Moderate comments:

1- There is an excellent matching between the varve count with the 2 dated wood samples. However, there is almost 6000 years (first ~360 cm) without chronological constraint. Given that many varves are qualified as 'unclear' from 130 cm to ~270 cm of the composite depth, perhaps some other dating techniques could be added such as paleomag, OSL, 14C, etc. I would encourage the authors to at least comment on this.

2- Have you used any particular software to count the varves, please provide what you used.

3- The names of the cores and their depth are indicated in Fig.3. However, it is unclear in my opinion which cores were used for the composite. I assume A1o, and some part of the A3o, A3u... In brief how much sediment was used from each core sections?

4- Fig. 1 : Have you obtained several (7) gravity cores that are not in the same location of the composite core?

5- Solar activity: Lines 414-416: Raspopov et al., (2008) use a 100-300 year band-pass filter and find 'great correlation' with solar activity (inferred from 14C) from three locations or so, and with lags (as high as 150 years). One can do the same analysis with white noise and find similar correlation (for example see Turner et al. 2016: Solar

cycles or random processes?). But more importantly, they filter out (bandpass) the data which make any high correlation not surprising at all. The comparison of the tree-rings and 14C prod rate (Fig. 1; Raspopov et al., 2008) without filtering is not very convincing either. Finally, they don't use the actual instrumental sunspots data spanning the past ∼300 years to compare with their tree-ring records, which is a little bit curious. To be honest, I don't reject the influence of solar forcing on regional climate, but based on this paper, it does not help your interpretation of the connection between solar forcing and your site.

5b: Lines 414: "which show decadal-to centennial periodicities". The authors refer to Fig. 4 LZ II. This is an image; hard to see any decadal-to centennial periodicities. Can you make spectral analysis of these layers characterizing lithozone II to prove these periodicities? It could be challenging without i.e. $\mu$-XRF data.

6- In the text the authors use AD, please add AD/BC in your plots.

Minor comments :

Lines 37-38 : Why Lake Telmen is varved ∼1940-2013? Human influence (N & P) in the watershed? If so, this is not the case for your site?

Figures 5 and 6 : add error bars on CRS/CIC model

Lines : 164-233-763 : change centimetre to centimeter

Figure 1: should add labelling to isobaths.

Line 301: laminar denudation: please describe this.

Line 461: Why such an increase of precipitation at AD 1150? MCA? However, it seems to last until recent, so occurring in the LIA as well. A change in boundary conditions in the watershed? High-resolution grain-size analysis could shed some light about this.

Turner, T. Edward, et al. "Solar cycles or random processes? Evaluating solar variability in Holocene climate records." Scientific reports 6.1 (2016): 1-15.

---

## Referee Comment (RC3) · Anonymous Referee #3 · 28 Feb 2020

This paper presents an annually-laminated sediment record from Lake Chatyr Kol. The paper focusses on establishing a process model for varve formation and then explaining how the floating varve chronology is 'fixed' to an absolute timescale. The paper is succinct and achieves the two aims that are set out at the end of the Introduction section. There is plentiful data that describes and explains the different microfacies that are observed in 6.235 m core sequence, detailed explanation of the generation of error from the varve counts and the use of 210Pb and radiocarbon dates. The radiocarbon dating programme identifies a significant reservoir effect that decreases in magnitude toward the younger sediments and explanations for the likely causes of this are included in the later section 5. Whilst the palaeoenvironmental interpretation in

[Figure]

Section 5 is limited, and concentrates on an initial interpretation of the different microfacies, it would appear deliberate in that the authors wish to establish the process model for varve formation and also the chronology development in this paper, before exploring the palaeoenvironmental significance of the sequence in later publications. Consequently, the current focus of the paper is appropriate and meets the scope of this journal. There are ways of possibly improving the manuscript to aid the readers understanding of the sequence and also make the varve microfacies sections a little easier to follow.

1. There are several points in Section 5 (eg. Line 298; Line 401; Line 435) that refer to the role of glacial meltwater in the supply of detrital clastic material into the basin and their presence in the varve microfacies. Is it possible to include the area that is/was glaciated in Figure 1? There is no mention of this in Section 2 (Study Site) other than meltwater run-off and it is not clear if this is from a glaciated catchment. A little more detail on this would be helpful to the reader. Also permafrost thaw is considered a possible contributor to older carbon in the lake water to explain the reservoir bu this is not described within the site context. Could this also be included in the site context?

2. Microfacies section – the introduction to section 4.2 might be considered contradictory in that Line 166 states ' consists of mainly clastic lamination' but Figure 4 has clastic material in all of the microfacies. Later in the paragraph, it is stated that 'subtypes were named according to the order of their dominant contents', which has two occasions where either organic or calcitic laminations dominate the microfacies making the earlier statement invalid. If the first sentence said 'Clastic material is present in all of the macroscopically visible laminations below 63.0 cm depth, and intercalates with calcitic, aragonitic and organic sublayers that build-up cyclic successions. And then the final three sentences can remain and it is a truer reflection of the microfacies. Also it could be useful to state how the subtypes are named according to the their dominant contents ( I assume that the dominant component comes first?).

3. Figure 4 is good at showing the broader differences in the microfacies in each of

the LZ's. However, the detail in the schematic (varve depositional model) is difficult to evaluate within the images from the thin sections at their current magnification. Could a higher magnification image that reflected more closely the elements shown in the schematic also be included? Also a key for the symbols in the schematic is necessary, and I note that there is no obvious winter layer detected in the clastic-diatom and clastic organic/clastic aragonitic microfacies. Related to this, in the text is 'section 4.2.1 Clastic-organic laminae' the lower schematic or the upper schematic? What is the difference between these two? It appears to be the aragonite and this is what is identified in the text, but the clastic-organic coming first in the Figure confuses this distinction. It would also be helpful that the order that is in the text was followed by the order in the figure to remove this confusion.

4. Section 5.4 provides a nice explanation of the broad environmental changes that lead to variations in the microfacies through the sequence. A criticism is that it is difficult to evaluate the thickness data of the different microfacies against the text, which starts by describing the frequency of the different microfacies in each of the Lithozones. A suggestion that could help the reader and also highlight the differences in microfacies that are observed between the LZ's would be to include on Figure 9 some percentage bar charts that collate the relative proportion of the different microfacies in each LZ. Such that for LZ I with clas-org 57%, clas-calc 29% and clas-arag 14%, clas-dia 0%, org-clas 0% calc-clas 0%. Then using the same order for the microfacies there could be a barchart for LZ II, LZ III etc and then if aligned vertically the reader could draw a direct comparison between LZ's seeing the changes through the sequence. This could be a column on the righthand side of the current Figure. It would also be useful to arrange the thickness graphs for each of the microfacies in the same order as their description in Figure 4 and in the text of Section 4.

Technical Corrections: Throughout the manuscript superscript is used inconsistently when it should be used e.g. for 14C. Lead-210 is used interchangeably with 210Pb , and cm-1 should be cm-1.

Spaces should be included between ages and the $\pm$ symbol.

Line 39 – states '…..which cover approximately 7,100 cal years BP….' , is that the duration of the record or the base of the sequence is dated using varves to 7,100 cal years BP. Is this also the case for Lake Sugan and is this also in cal yrs BP?.

Line 44 – remove 'n'

Line 92 – where they are archived in a cold store at 4° C

Line 97 – remove 'continuously' and put 'Continuous' at the start of the sentence.

Line 138- should a value for keV be included after 5.9%?

Line 175 – is there an image that illustrates how it is possible to distinguish between detrital and endogenic calcite?

Line 301- I was unclear on 'laminar denudation' is that erosion of the lamination?

Line 311 – replace 'overserved' with 'observed'?

Line 343-344 – I was not clear on the meaning of 'for each individual thin section comprising 324 and 13 years varve. My assumption is that this is the range, or maximum and minimum, in total number of varves observed on a single 10 cm thin section. However, I may have misread this.

Line 474 – unclear on the meaning of 'robust fundament'. Do you mean 'This robust chronology is fundamental for further detailed palaeoenvironmental…….'?

Line 479 – I assume that the increased windiness enable increased mixing of the lake waters and CO2 exchange with the atmosphere. Perhaps be explicit here.

Line 480 replace 'which allowed developing' with ….'that allowed the development of seasonal deposition models

---

## Author Comment (AC1) · 20 Mar 2020

We appreciate the constructive and very helpful comments on of anonymous referee #1 and we addressed suggestions in our revision. However, we relinquish from including more information about regional climate changes in Central Asia. It is the explicit aim of this study to develop a robust chronology for the Chatyr Kul sediment record based predominantly on varve counting. This is the first varve chronology for a Central Asian lake sediment record and detailed analyses of seasonal sedimentation processes were required to prove the existence of annual laminations. This chronology further enabled us through comparison with radiocarbon dating on aquatic matter for the first time (i) to

quantify reservoir ages in a Central Asian lake record and (ii) to demonstrate changes of reservoir ages throughout the Holocene. Our results are of general interest for lake sediment dating in this region in settings where only radiocarbon dating is possible. Hence, we consider our chronological data as valuable stand-alone results and, therefore, have chosen Geochronology as a target journal. We explicitly disclaim from discussing climate changes because this would have been far beyond the scope of this paper and, in contrast, even would have blurred its focus. We have clarified the focus of this paper in the manuscript (lines 51-54 in the revised manuscript). The results of this study form a robust chronological base for future environment and climate reconstructions from the Chatyr Kol sediments.

Specific, minor suggestions:

Referee comment: Lines 14-17. These first two sentences of the abstract appear to be repetitive and could be combined in shortened form.

Author's response:corrected

Author's change in the manuscript: Microfacies analysis of a sediment record from Lake Chatyr Kol (Kyrgyz Republic) reveals the presence of seasonal laminae (varves) from the sediment basis dated at 11,619 $\pm$ 603 years BP up to ~360 $\pm$ 40 years BP. The Chatvd19 floating varve chronology relies on replicate varve counts on overlapping petrographic thin sections with an uncertainty of $\pm$ 5 %.

Referee comment: Line 44: Typo – "n" to be removed

Author's response:corrected

Author's change in the manuscript: . . .2) human influence (Boomer et al., 2000; Mathis et al., 2014; Schröter et al., 2019 in review),. . .

Referee comment: Lines 53-54: This information should be included into supplements. The aim of these projects is obviously the reconstruction of Holocene climate and so more information on this should be provided in the paper.

Author's response: This was indeed misleading since we did not clearly distinguish between the overall project goals on climate reconstruction (which involves also other research groups) and the scope of this study, i.e. the construction of a robust age model for the entire project team. We have clarified the goal of this study in the revised version (lines 51-54).

Author's change in the manuscript: The sediment record from Lake Chatyr Kol is the first varved record from CA covering most of the Holocene and the main goal of this study is to establish a robust age model through an integrated dating approach primarily based on varve counting. Varve counting requires an in-depth understanding of seasonal deposition of all varve types occurring in the sediment record.

Referee comment: Line 62: remove dash

Author's response:corrected

Author's change in the manuscript: Geologically, the surrounding mountain ranges belong to Silurian to Carboniferous sedimentary-volcanogenic complexes of marine-continental collision zones, consisting of limestones and dolomites, that crop out directly along the northern lake shore, as well as siliceous rocks, shales and scattered Permian granites that crop out in the south and north-east (Academy of Science of the Kyrgyz SSR, 1987).

Referee comment: Line 70: delete amounts

Author's response:corrected

Author's change in the manuscript: Mean annual precipitation is ∼275 mm/a as indicated by Aizen's (2001) evaluation and spatial averaging of annual 70 precipitation amounts of historical records published by Hydrometeo (Reference Book of Climate USSR, Kyrgyz SSR, 1988).

Referee comment: Line 133: How long was the in-growth time (check also spelling to change to "in-growth-time". The term photo peak activity sounds incorrect to me

and should be replaced with a more appropriate term. It is the gamma energy that is recorded in the gamma spectrometry.

Author's response:We changed "photo peak activities" to gamma energies and changed ingrowth-time to in-growth-time.

Author's change in the manuscript: After sufficient in-growth-time, the gamma energies of 210Pb (T1/2= 22 a) and 214Pb (T1/2= 26.8 min), which is a daughter nuclide of 222Rn (T1/2= 3.8 d), were measured at 46.54, 295.24 and 351.93 keV. Hardware control, data storage, and spectrum analysis were realized with the software Genie 2000 (Canberra Industries). Measurements were taken out for 1.5 to 7 days (Suppl. Tab. 1).

Referee comment: Line 139: lab-internal – please note which lab, and where these samples were analyzed.

Author's response:Information has been added.

Author's change in the manuscript: For this purpose, the Kryal$^{©}$ tubes were placed into two well-type germanium detectors G1 and G2 (Canberra Industries) located in a basement lab of a concrete building at GFZ Potsdam which is actively ventilated (Schettler et al., 2006).

Referee comment: Line 167: use a, b, c to refer to each group of laminae more easily and use this instead of "LZ + number" in the references to Fig 4 throughout the text

Author's response:corrected

Author's change in the manuscript: Figure captions and figure 4 have been changed accordingly, also in the text.

Referee comment: Line 204: Add current institute/university of Ms Schwarz within brackets as well

Author's response:Information has been added.

Author's change in the manuscript: The third sublayer is formed by diatom blooms exclusively consisting of the planktic diatom species Cyclotella choctawhatcheeana (pers. comm. Anja Schwarz, TU Braunschweig) (Fig. 4.1 b upper part).

Referee comment: Line 231: Add picture of homogeneous sediment to Fig 4 as well to see how it com-pares to the varved intervals. In particular, this is useful to show the faint, discontinuous laminae in the uppermost cm

Author's response: agreed.

Author's change in the manuscript: Pictures of homogenous sediments with faint lami-nae have been added to figure 4.

Referee comment: Lines 269-270: Is this assumption justified? +/- 40 years BP uncer-tainty could be higher or lower? Why is it not possible to more precise that this? On the basis of the data presented, I would be surprised if the error is as high as 40 year

Author's response: We consider an uncertainty of +/- 40 years at the anchor point as justified for the non-varved interval given the counting uncertainty of ca 5% in laminated sections. We agree to the reviewer that this error might be over-estimated but we prefer to provide a conservative estimate and clarified in the revised text that this uncertainty is considered a maximum range.

Author's change in the manuscript: We assume a conservative uncertainty of ca. 10% as a maximum error for our interpolation.

Referee comment: Line 386: change to effect

Author's response: done.

Author's change in the manuscript: The abrupt decrease of the reservoir effect after ∼AD 1150, despite an increase in detrital carbonate supply (Sect. 5.3.5, Fig. 9) might be related to the silting up of the basin leading to a shallower water depth, which is more susceptible to water circulation and an enhanced atmospheric $CO_2$ exchange

(c.f. Geyh et al., 1997).

Referee comment: Line 491 (data availability statement): Please add the data into this database during the review process, so you can include the doi of the dataset in this statement. I think itis very important to add the doi to the final paper, so the future reader can access the datasets easily

Author's response: done. We further added a link to the newly established VARDA database

Author's change in the manuscript: https://doi.pangaea.de/10.1594/PANGAEA.909981 https://varve.gfz-potsdam.de

––––––––––––––––––––––––––––––––––––

[Figure]

[Figure]

**Fig. 1.** Figure 4:1 Thin section pictures of different lamination (varve) types in cross-polarized (left) and plane polarized (right) light of the different lithozones LZ I to LZ V. a) Clastic-organic laminati

---

## Author Comment (AC2) · 20 Mar 2020

We thank reviewer #3 for his appreciation of our work and his valuable suggestion that we include in our revision.

Referee comment: 1. There are several points in Section 5 (eg. Line 298; Line 401; Line 435) that refer to the role of glacial meltwater in the supply of detrital clastic material into the basin and their presence in the varve microfacies. Is it possible to include the area that is/was glaciated in Figure 1? There is no mention of this in Section 2 (Study Site) other than meltwater run-off and it is not clear if this is from a glaciated catchment. A little more detail on this would be helpful to the reader. Also permafrost

thaw is considered a possible contributor to older carbon in the lake water to explain the reservoir but this is not described within the site context. Could this also be included in the site context?

Author's response: The area displayed in figure 1 does not exhibit recent glaciers, but several glaciers exist further north-east on the central At Bashy range (Narama et al., 2007) at a level above ~4000 m a.s.l. and some of them drain into the Chatyr Kol via the Kegagyr River. Recent glaciers are also located on the western Torugat range (not shown in Fig.1) but these glaciers do not drain into Chatyr Kol. We clarify glacier information in the chapter "study site". We further extend the formulation 'glacier and snow meltwater' because runoff includes also seasonal snowmelt. Finally, a photo of field observations displaying thermokarst/permafrost thawing structures observed at the Maloye Lake in < 2 km distance from Chatyr Kol is added.

Author's change in manuscript: Line 65-68: The modern lake, which has a maximum length of 23 km, a width of 10 km and a maximum depth of 20 m in its western-central part, is endorheic and separated from the neighboring Arpa river basin in the north-west by a moraine (Shnitnikov, 1978). The moraine originated from glacial advances of unknown age from the western Torugat range. Present day glaciers on the Torugat and At Bashy range exist above ~4000 m a.s.l. but only some of the At Bashy glaciers drain into Chatyr Kol via the Kegagyr River. The lake is moreover fed by convective rainfall events in summer (Aizen et al., 2001). A shallow watershed hinders outflow to the east.

Line 83-84: The permafrost level is located at a depth of 2.5-3 m in the littoral coast zones and the lake is covered by ice from October to April (Shnitnikov, 1978). Modern permafrost thawing results in instable shores visible at the Maloye lake located < 2 km to the South of Chatyr Kol(Fig. 1 Photo) and the development of small ponds on the shallow south-western shore of this lake and lake Chatyr Kol during the summer season.
Referee comment: 2. Microfacies section – the introduction to section 4.2 might be considered contradictory in that Line 166 states 'consists of mainly clastic lamination' but Figure 4 has clastic material in all of the microfacies. Later in the paragraph, it is stated that 'sub-types were named according to the order of their dominant contents', which has two occasions where either organic or calcitic laminations dominate the microfacies making the earlier statement invalid. If the first sentence said 'Clastic material is present in all of the macroscopically visible laminations below 63.0 cm depth, and intercalates with calcitic, aragonitic and organic sublayers that build-up cyclic successions. And then the final three sentences can remain and it is a truer reflection of the microfacies. Also it could be useful to state how the subtypes are named according to the their dominant contents ( I assume that the dominant component comes first?).

Author's response: The referee is right that our formulation was not sufficiently clear and we will change the introduction to chapter 4.2 accordingly. We only want to point out, that (i) all six varve types include a clastic sublayer and that (ii) the differentiation of varve types relies on the composition of the alternating sublayers and the dominance of sublayers within a varve cycle. The latter criterium is used for varve type names. For example, the clastic-organic varve type is characterized by the dominance of the clastic sublayer, while in the organic-clastic varve type the organic sublayer prevails. The varve types names are not related to the order of sublayer succession within the varves. In addition to the revision of the introduction of chapter 4.2, we re-name the subchapters from 'Clastic-organic laminae' to Clastic-organic type' etc. to clarify that we are not presenting individual sublayers but varve types.

Author's change in manuscript: Line 166: Microscopic sediment analysis revealed, that clastic sublayers are present throughout the finely laminated sediments below 63.0 cm depth (Fig. 4.1). These clastic sublayers are variably intercalated with calcitic, aragonitic and organic sublayers and thus form different types of cyclic successions. In total, we classified six different types of sublayer successions as described below. The name for these types reflects the dominant sublayer for each of the six types.

For example, the 'clastic-organic type' is characterized by the dominance of clastic sublayers, while in the organic-clastic type organic sublayers dominate. The names are not related to the order of sublayer succession within each type.

Referee comment: 3. Figure 4 is good at showing the broader differences in the microfacies in each of the LZ's. However, the detail in the schematic (varve depositional model) is difficult to evaluate within the images from the thin sections at their current magnification. Could a higher magnification image that reflected more closely the elements shown in the schematic also be included? Also a key for the symbols in the schematic is necessary, and I note that there is no obvious winter layer detected in the clastic-diatom and clastic organic/clastic aragonitic microfacies. Related to this, in the text is 'section 4.2.1Clastic-organic laminae' the lower schematic or the upper schematic? What is the difference between these two? It appears to be the aragonite and this is what is identified in the text, but the clastic-organic coming first in the Figure confuses this distinction. It would also be helpful that the order that is in the text was followed by the order in the figure to remove this confusion.

Author's response: We include additional microscopic images at higher magnification in the supplement. (Suppl. Fig. 2) In addition, we include $\mu$XRF element mapping of selected thin sections in order to better visualize the varve facies (new Fig. 4.2). Keys/Legends will be added for the used symbols in figure 4. Winter layers in clastic-organic/clastic-aragonitic varve deposition model are added in fig. 4. The reviewer is also right that the relation of schematics in figure 4 to section 4.2.1 is not clear. Therefore, we modified fig. 4 accordingly.

Author's change in manuscript: We will change the order of the sub chapters in 4.2 according to figure 4.1 and revise the varve schemata label in fig 4.1.

Referee comment: 4. Section 5.4 provides a nice explanation of the broad environmental changes that lead to variations in the microfacies through the sequence. A criticism is that it is difficult to evaluate the thickness data of the different microfacies

against the text, which starts by describing the frequency of the different microfacies in each of the Lithozones. A suggestion that could help the reader and also highlight the differences in microfacies that are observed between the LZ's would be to include on Figure 9 some percentage bar charts that collate the relative proportion of the different microfacies in each LZ. Such that for LZ I with clas-org 57%, clas-calc 29% and clas-arag 14%, clas-dia 0%,org-clas 0% calc-clas 0%. Then using the same order for the microfacies there could be a bar chart for LZ II, LZ III etc and then if aligned vertically the reader could draw a direct comparison between LZ's seeing the changes through the sequence. This could be a column on the right hand side of the current Figure. It would also be useful to arrange the thickness graphs for each of the microfacies in the same order as their description in Figure 4 and in the text of Section 4.

Author's response: The referee is right and changes in figure 9 have been made.

Author's change in manuscript: Instead of using bar plots, pie charts are displayed for each lithozone in figure 9. The thickness graphs in figure 9 are ordered according to the order in the text in chapter 4.2.

Technical corrections:

Referee comment: Throughout the manuscript superscript is used inconsistently when it should be used e.g. for 14C. Lead-210 is used interchangeably with 210Pb, and cm-1 should be cm-1. Spaces should be included between ages and the $\pm$ symbol.

Author's response: agreed. The text will be changed accordingly.

Referee comment: Line 39 – states '.....which cover approximately 7,100 cal years BP...' , is that the duration of the record or the base of the sequence is dated using varves to 7,100 cal years BP. Is this also the case for Lake Sugan and is this also in cal yrs BP?

Author's response: The lake sediment record of Lake Telmen extends back to 7,100 cal years BP according to AMS radiocarbon dating of bulk sediment and pollen extracts.

[Figure]

Varves in the Telmen record were only found in sediments younger than 4,390 cal yr BP (AMS) and have not been counted because varve preservation is reported as discontinuous. Instead, Peck et al. (2012) extrapolated average couplet thickness for an indirect varve age estimate. For Lake Sugan, the authors counted laminae couplets on digital images of split core surfaces and compared these with 210Pb dating (CRS). The ages were not provided as cal yrs BP.

Author's change in manuscript: In Kyrgystan, varves have been only reported from Lake Sary Chelek for the short time interval from ∼1940's to 2013 (Lauterbach et al., 2019). Other varved records in the wider region are Lake Telmen in northern Mongolia which exhibits discontinuous varved intervals during the last ca. 4,390 cal years BP (Peck, 2002) and Lake Sugan in north western China covering the last ∼2,670 years BP (Zhou et al., 2007).

Referee comment: Line 44 – remove 'n' Author's response: Will be removed.

Referee comment: Line 92 – where they are archived in a cold store at 4âŮęC

Author's response: agreed

Author's change in manuscript: All cores were opened, split and photographed at GFZ Potsdam, where they are archived in a cold store at 4°C.

Referee comment: Line 97 – remove 'continuously' and put 'Continuous' at the start of the sentence.

Author's response: agreed

Author's change in manuscript: Continuous 10-cm-long sediment slabs with an overlap of 2 cm were taken from the whole composite profile to prepare large-scale petrographic thin sections.

Referee comment: Line 138- should a value for keV be included after 5.9%?

Author's response: The keV for 210Pb and 137Cs were mentioned already earlier in

the text in line 134-135.

Author's change in manuscript: none

Referee comment: Line 175 – is there an image that illustrates how it is possible to distinguish between detrital and endogenic calcite?

Author's response: agreed

Author's change in manuscript: An image (new Fig. 4.2) (microscopic pictures) illustrating the differences is added to the manuscript.

Referee comment: Line 301- I was unclear on 'laminar denudation' is that erosion of the lamination?

Author's response: By "laminar denudation" we mean the superficial runoff associated with catchment runoff through the activation of widely dispersed smaller tributaries. We changed the wording from "laminar denudation" to "surface runoff" and clarify the meaning.

Author's change in manuscript: Runoff with suspended sediment load is then likely directed through the Kegagyr River in the east but may also be the result of surface runoff through the activation of several widely distributed smaller tributaries in the catchment.

Referee comment: Line 311 – replace 'overserved' with 'observed'?

Author's response: agreed.

Author's change in manuscript: Aragonite precipitates were only observed in the intervals between 600.0-605.0 and 609.0-616.0 cm composite depth.

Referee comment: Line 343-344 – I was not clear on the meaning of 'for each individual thin section comprising 324 and 13 years varve. My assumption is that this is the range, or maximum and minimum, in total number of varves observed on a single 10 cm thin section. However, I may have misread this.

Author's response: This is correct, this refers to maximum and minimum number of varves within individual thin sections.

Author's change in manuscript: For the floating varve chronology we therefore compare the results for each individual thin section comprising between a maximum of 324 (506.8-497.6 cm) and a minimum of 13 (varves) (65.4-63.0 cm) (Fig. 5a, Fig. 8).

Referee comment: Line 474 – unclear on the meaning of 'robust fundament'. Do you mean 'This robust chronology is fundamental for further detailed palaeoenvironmental.......'?

Author's response: Yes.

Author's change in manuscript: This robust chronology forms the base for further detailed palaeoenvironmental and palaeoclimatic reconstructions.

Referee comment: Line 479 – I assume that the increased windiness enable increased mixing of the lake waters and $CO_2$ exchange with the atmosphere. Perhaps be explicit here.

Author's response: agreed.

Author's change in manuscript: Lower reservoir ages of ~1000 years and less in the late Holocene might be related to enhanced atmospheric $CO_2$ exchange when the lake was shallower due to silting-up of the lake basin and/or increased windiness inducing increased water column mixing favoring $CO_2$ exchange with the atmosphere.

Referee comment: Line 480 replace 'which allowed developing' with....'that allowed the development of seasonal deposition models

Author's response: agreed.

Author's change in manuscript: The construction of the varve-based chronology was only possible through detailed micro-facies analyses of the entire sediment sequence in overlapping thin sections that allowed the development of seasonal deposition models

for all observed types of fine laminations.

References Narama, C., Kääb, A., Duishonakunov, M., & Abdrakhmatov, K. (2010). Spatial variability of recent glacier area changes in the Tien Shan Mountains, Central Asia, using Corona ($\sim$ 1970), Landsat ($\sim$ 2000), and ALOS ($\sim$ 2007) satellite data. Global and Planetary Change, 71(1-2), 42-54.

Figure 1 caption: Figure 1: Location of Lake Chatyr Kul, the composite profile (red dot) and the gravity cores (yellow dots). The orange square marks the location of 14C dated leaves (Poz-109830, Tab. 2) found in the top of a mid-Holocene-shoreline at $\sim$3540 m a.s.l. The relief map of Kyrgyzstan relies on the CGIAR-CSI SRTM 90m (3 arcsec) digital elevation data (Version 4) of the NASA Shuttle Radar Topography Mission (Jarvis, 2008). The figure was modified from Lauterbach et al. (2014). Photo of instable shores (white arrow) of Maloye lake.
* * *
[Figure]

[Figure]

**Fig. 1.** Figure 1: Location of Lake Chatyr Kul, the composite profile (red dot) and the gravity cores (yellow dots). The orange square marks the location of 14C dated leaves (Poz-109830, Tab. 2) found in the t

---

## Author Comment (AC3) · 20 Mar 2020

Referee comment: Although I have no major comment on the central topic of this paper (that is suitable for the journal), i.e. the chronology, I am puzzled why there is no $\mu$-XRF data (e.g. Itrax) shown in your study. For example, the authors describe periods of prevailing anoxic bottom water conditions, calcitic materials/diatoms, coarse vs finer sediments, etc. In my opinion, it would be very helpful to show $\mu$-XRF elements (and elemental ratios) to support your visual microscopic analysis. Have you made such analysis (XRF)? If you are to interpret the paleoenvironments from this site in the paper, I think that would be very valuable.

[Figure]

Author's response: We appreciate the constructive comments and suggestions of referee #2 and we now included XRF mapping to strengthen detailed microfacies analysis presented in our MS. The main focus of the MS is a detailed microfacies analysis for 1) the development of process-based deposition models and 2) the establishment of detailed varve-based chronology. For these purposes, detailed microfacies analyses is crucial and allows to distinguish, for example, detrital from endogenic calcite and detrital quartz from diatom SiO2. Such differentiations are not possible using $\mu$-XRF scanning as element abundances, as these sediment fractions are geochemically identical and it is not apparent from, for example, relative variations of calcium and silicon respectively. However, we do agree that XRF element scanning is a powerful way to complement and support visual microfacies observations. Therefore, XRF scanning maps of sediment blocks are used, which are the equivalent of the thin sections used for the microfacies analyses. These new mapping results for selected intervals with characteristic varve types confirm the occurrence of both detrital and endogenic calcite. These data are presented in a new Figure 4.2 and in the main text (new chapters 3.3 and 4.4 $\mu$XRF element mapping, discussion within chapters 5.4.2 and 5.4.3) of the manuscript. To our opinion, XRF mapping results are most suitable for a precise linking of sediment compositions and microscopic observations. We add Rik Tjallingii as a co-author because he conducted the $\mu$XRF element mapping and helped with revising the manuscript.

Moderate comments:

Referee comment:

1.There is an excellent matching between the varve counts with the 2 dated wood samples. However, there is almost 6000 years (firstâĹij360 cm) without chronological constraint. Given that many varves are qualified as 'unclear' from 130 cm toâĹij270 cm of the composite depth, perhaps some other dating techniques could be added such as paleomag, OSL, 14C, etc. I would encourage the authors to at least comment on this.

Author's response: The reviewer is right that the age uncertainties are higher in this interval of less well preserved varves which we have addressed by allocating higher uncertainty ranges. Nevertheless, varves in this interval can still be counted and represent the only applicable dating method. We have sieved the entire interval in order to find terrestrial plant remains for radiocarbon dating but, unfortunately, without any success. All five 14C dates obtained from this interval are from aquatic material and thus revealed too old ages due to reservoir effects. Independently paleosecular variation (PSV) records in this region are only available from Lake Issyk Kol, Lake Baikal and Lake Aslikul. However, these records suffer from dating problems and show significant temporal offsets before 500 AD between the records and to global geomagnetic field models (Gómez‐Paccard, 2012) so that they cannot be used for the Chatyr Kol chronology. OSL dating is also not applicable because the Chatyr Kol sediments are mainly composed of materials not suitable for reliable luminescence dating including carbonates, organics and non-aeolian siliciclastic.

Author's change in manuscript: We have changed 'unclear' to 'less well preserved' varves (chapter 3.2).

Referee comment: 2. Have you used any particular software to count the varves, please provide what you used.

Author's response: No software was used for varve counting. Counting was exclusively performed on the Axioplan microscope using different magnifications and based on expert knowledge.

Author's change in manuscript: none

Referee comment: 3. The names of the cores and their depth are indicated in Fig.3. However, it is unclear in my opinion which cores were used for the composite. I assume A1o, and some part of the A3o, A3u...In brief how much sediment was used from each core sections?

Author's response: The core sections used for the composite profile are colored in grey, as indicated by the legend.

Author's change in manuscript: We add the depth sections of each core used for the composite profile in fig. 3.

Referee comment: 4. Fig. 1: Have you obtained several (7) gravity cores that are not in the same location of the composite core?

Author's response: As indicated in Figure 1, the gravity cores 3, 5, 6 and 7 were obtained close to the composite core location while gravity cores 1 and 2 were recovered about 1-1.5 km further north-east and number 4 was recovered $\sim$ 10 km further east in the shallow eastern lake basin.

Author's change in manuscript: none

Referee comment: 5. Solar activity: Lines 414-416: Raspopov et al., (2008) use a 100-300 year band-pass filter and find 'great correlation' with solar activity (inferred from 14C) from three locations or so, and with lags (as high as 150 years). One can do the same analysis with white noise and find similar correlation (for example see Turner et al. 2016: Solar cycles or random processes?). But more importantly, they filter out (bandpass) the data which make any high correlation not surprising at all. The comparison of the tree-rings and 14C prod rate (Fig. 1; Raspopov et al., 2008) without filtering is not very convincing either. Finally, they don't use the actual instrumental sunspots data spanning the pastâĹij300 years to compare with their tree-ring records, which is a little bit curious. To be honest, I don't reject the influence of solar forcing on regional climate, but based on this paper, it does not help your interpretation of the connection between solar forcing and your site. 5b: Lines 414: "which show decadal-to centennial periodicities". The authors refer to Fig. 4 LZ II. This is an image; hard to see any decadal-to centennial periodicities. Can you make spectral analysis of these layers characterizing lithozone II to prove these periodicities? It could be challenging without i.e. $\mu$-XRF data.

Author's response: We agree to the reviewer and delete the discussion on solar cycles. We restrain to a pure description of the observed intercalations and point out that it remains unclear if they are related to external triggers or random processes citing the reference suggested by the reviewer (Turner et al., 2016). We also change the term "periodicities" to intercalating /recurring patterns.

5b. As we agree to the reviewer concerning the occurrence of petrographic periodicities and avoid this term in the revised manuscript, there is no further need for spectral analyses. The deposition of differing varve types showing the decadel-cenntennial intercalating pattern is purely based on varve counting. To better visualize the intercalations we add a figure linking sediment compositions and microfacies observations using XRF mapping of the varve type distribution for selected intervals of LZ II and LZ III (new Fig. 4.2).

Author's change in manuscript: The causes for these clear intercalations remain speculative and include either external (climatic) triggers or unknown lake-internal or sedimentation variability. (Turner et al., 2016 and reference herein). Fig. 4.2 ($\mu$XRF mapping) has been added.

Referee comment: 6. In the text the authors use AD, please add AD/BC in your plots.

Author's response:agreed.

Author's change in manuscript: An axis of AD/BC ages is added to the plots.

Minor comments:

Referee comment: Lines 37-38: Why Lake Telmen is varvedâĹij1940-2013? Human influence (N & P) in the watershed? If so, this is not the case for your site?

Author's response: This was misunderstood by the reviewer because the varve record from 1940 – 2013 is from Lake Sary Chelek in Kyrgystan and not from Lake Telmen. From Lake Telmen discontinuous varved intervals are reported for the time period from 4,390 cal years BP on (Peck, 2002). We clarify the sentences about other regional

varve records in the text.

Author's change in manuscript: In Kyrgyzstan, varves have so far been only reported from Lake Sary Chelek (Kyrgyzstan) for the short time interval from ∼1940's to 2013 (Lauterbach et al., 2019). Other varved records in the larger region are Lake Telmen in northern Mongolia which exhibits discontinuously varved intervals from approximately 4,390 cal years BP (Peck, 2002) and Lake Sugan in north western China covering the last ∼2,670 years BP (Zhou et al., 2007).

Referee comment: Figures 5 and 6: add error bars on CRS/CIC model

Author's response:done.

Author's change in manuscript: Error bars added in Figures 5 & 6.

Referee comment: Lines : 164-233-763 : change centimetre to centimeter

Author's response:agreed.

Author's change in manuscript: Line 164: The uppermost centimeter is enriched in calcite and exhibits greyish faint laminations. Line 233: Faint and discontinuous calcite laminae occur in the uppermost centimeter (Fig. 4.1 f). Line: 763: (Fig.6) Core pictures of the upper part of the composite profile CHAT12 (right) and the gravity core SC17_7 (left) illustrate the facies change to calcite-enriched sediments in the uppermost centimeter.

Referee comment: Figure 1: should add labelling to isobaths.

Author's response:agreed.

Author's change in manuscript: Isobaths are labelled in figure 1.

Referee comment: Line 301: laminar denudation: please describe this.

Author's response: By "laminar denudation" we mean the superficial catchment runoff probably associated with an activation of widely dispersed smaller tributaries during

precipitation events. We will clarify this in the text.

Author's change in manuscript: Runoff with suspended sediment load is then likely directed through the Kegagyr River in the east but may also be the result of surface runoff through the activation of several and widely distributed smaller tributaries in the catchment.

Referee comment: Line 461: Why such an increase of precipitation at AD 1150? MCA? However, it seems to last until recent, so occurring in the LIA as well. A change in boundary conditions in the watershed? High-resolution grain-size analysis could shed some light about this.

Author's response: It is correct that the onset of additional detrital sub-layers during summer started at AD 1150 but it lasted not until recent but until ca. AD 1730, about the time when varve preservation became poor and finally ceased. We do not know the reason for this increase in summer runoff events and can only state that there is no coincidence with climatic periods reported from other records (MCA, LIA). We did not carry out grain size analysis because our continuous microfacies analyses does not reveal any significant shift in grain size. Therefore, changes in boundary conditions in the catchment appear unlikely. We have revised and clarified the text accordingly.

Author's change in the manuscript: Chapter 5.4.5revised Clastic-organic varves constitute 59 % of the observed varves in LZ V, clastic-calcitic varves 26 % and organic-clastic varves 15 %, the latter ceasing at 110.5 cm (AD 1260 $\pm$ 50). Varve microfacies changes abruptly at 130 cm depth or AD 1150 from the dominance of organic-clastic varves to dominating clastic-organic and clastic-calcitic varves. Within 5 years, varve thickness drastically increase from Ø 0.43 mm in LZ IV to Ø 1.52 mm in LZ V due to thicker summer sublayers. Thicker summer sublayers result from both thicker mixed sublayers rich in algae remains (Botryococcus, chrysophytes, diatoms) and additional late summer detrital sublayers (Fig. 4.1.e, Suppl. Fig. 2f). The increase in summer layer thickness, therefore, suggest both, higher lacustrine productivity and an increase

in summer runoff. However, the reasons for these changes remain elusive and a relation to known climatic periods like the Medieval Climate Anomaly and the Little Ice Age is not found. One might speculate that the frequent occurrence of late summer runoff layers either reflects convective rainfall events due to recycling of local moisture sources (Aizen et al., 2001), or changing atmospheric circulation regimes. Changes in boundary conditions in the catchment of the lake are unlikely since microfacies analyses does not show pronounced changes in grain size distribution of the detrital material. Human impact cannot fully be excluded but low indices of human and livestock fecal biomarkers (Schroeter et al., 2020) are an argument against major human impact. The presence of lake deposits at the northern and southern shores ca 1.5 - 1 m above present day lake level dated at AD 1420 $\pm$ 204, AD 1044 $\pm$ 160 and AD 858$\pm$ 166 (Shnitnikov, 1978) suggests that increased summer runoff might have resulted in a more positive water budget and lake level rise.

References

Gómez‐Paccard, M., Larrasoaña, J. C., Giralt, S., & Roberts, A. P. (2012). First paleomagnetic results of mid‐to late Holocene sediments from Lake Issyk‐Kul (Kyrgyzstan): Implications for paleosecular variation in central Asia. Geochemistry, Geophysics, Geosystems, 13(3).

Schroeter, N., Lauterbach, S., Stebich, M., Kalanke, J., Mingram, J., Yildiz, C., .Shouten, S. & Gleixner, G. (2020). Biomolecular evidence of early human occupation of a high-altitude site in Western Central Asia during the Holocene. Frontiers in Earth Science, 8, 20. https://doi.org/10.3389/feart.2020.00020
* * *